# Systematic identification of bacterial factors driving *Staphylococcus aureus* intracellular lifestyle in non-professional phagocytes

Ines Rodrigues Lopes[1,2], Laura Maria Alcantara[1,2], Maria Lopez-Bravo[3], Yi Liu [4], Gerald Larrouy-Maumus [4], Daniel Lopez [3], Miguel Mano [2,5] & Ana Eulalio [1,4]

*Staphylococcus aureus* is a major human pathogen responsible for severe infections. While traditionally described as extracellular, increasing evidence establishes *S. aureus* as a facultative intracellular pathogen. Intracellularity contributes to immune evasion, dissemination, and antibiotic failure. To identify bacterial factors critical for *S. aureus* invasion, intracellular replication, persistence, and host cytotoxicity, we screened a comprehensive collection of 1920 *S. aureus* mutants (Nebraska Transposon Mutant Library) in epithelial cells across five timepoints (0.5 to 48 hours post-infection). We identified 73 bacterial factors strongly modulating *S. aureus* intracellularity, including mutants displaying multiple phenotypes. Most of these factors have not been linked to intracellular lifestyle. Among these, we characterized the nicotina-midase PncA as a novel regulator of the *agr* system via redox state modulation, strongly impacting virulence. This study provides a systematic analysis of *S. aureus* factors critical for intracellular lifestyle, with implications for the development of antimicrobial strategies targeting this resilient bacterial population.

S*taphylococcus aureus* (*S. aureus*) is a Gram-positive bacterium that frequently colonizes the human skin and anterior nares as part of the normal microflora[1]. Somehow paradoxically, *S. aureus* is also an opportunistic and very successful human pathogen, causing a broad spectrum of diseases, associated with more than 1 million deaths/year[2]. Moreover, infections caused by methicillin-resistant *S. aureus* (MRSA) strains are the second most common cause of death associated with bacterial antimicrobial resistance[3]. The success of *S. aureus* resides in its ability to escape the host immune response and to resist most conventional antimicrobial therapies using an array of mechanisms, including its capacity to reside inside host cells[4].

Compelling evidence demonstrates that *S. aureus* can invade, replicate, and persist within a variety of human professional and non-professional phagocytic cells, including neutrophils, macrophages, epithelial and endothelial cells, osteoblasts, keratinocytes, and fibroblasts[5,6]. Intracellular *S. aureus* has also been observed in various human tissues, including nasal polyps, tonsils, nasal tissue from healthy carriers and patients with recurrent sinusitis, and osteoblasts from patients with chronic osteomyelitis[6–9]. Supporting this, our recent work revealed that the intracellular lifestyle is prevalent among *S. aureus* clinical isolates from patients with bone/joint infections, bacteraemia, and infective endocarditis[10]. We identified distinct

[1]RNA & Infection Laboratory, Center for Neuroscience and Cell Biology (CNC-UC), Centre for Innovative Biomedicine and Biotechnology (CIBB), University of Coimbra, Coimbra, Portugal. [2]Functional Genomics and RNA-based Therapeutics Laboratory, Center for Neuroscience and Cell Biology (CNC-UC), Centre for Innovative Biomedicine and Biotechnology (CIBB), University of Coimbra, Coimbra, Portugal. [3]National Centre for Biotechnology, Spanish National Research Council (CNB-CSIC), Madrid, Spain. [4]Centre for Bacterial Resistance Biology (CBRB), Department of Life Sciences, Imperial College London, London, UK. [5]School of Cardiovascular and Metabolic Medicine & Sciences, King's College London, London, UK. ✉e-mail: miguel.mano@kcl.ac.uk; a.eulalio@imperial.ac.uk

intracellular profiles characterized by varying levels of bacterial replication, persistence, and host cell death.

The intracellular *S. aureus* reservoir has been proposed to contribute to chronic and relapsing infections, linked to evasion of host immune defences, bacterial dissemination, and failure of antibiotic treatments[4,6]. The latter is particularly relevant given the inefficacy of many of the antibiotics most commonly used to target intracellular pathogens, owing to their low cellular penetration and intracellular retention, inability to reach specific subcellular compartments where bacteria reside, and reduced activity due to pH and/or enzymatic inactivation[11,12]. Along this line, it has been shown that strategies targeting intracellular *S. aureus* (e.g., antibody-antibiotic conjugates, natural killer cell mimics) are more efficient in treating infections when compared to currently used antibiotics (e.g., vancomycin)[9,13].

Although several key determinants of *S. aureus* intracellular lifestyle have been identified, our understanding of the mechanisms underlying intracellular survival remains incomplete. *S. aureus* adhesion and uptake by non-professional phagocytes are mediated by several bacterial adhesins, which interact with extracellular matrix proteins or specific host receptors[14]. Once internalized into a vacuole, *S. aureus* can have diverse intracellular fates, including rapid clearance, replication, induction of host cell death, and/or persistence[5,6]. Escape from the bacteria-containing vacuole into the host cell cytoplasm was shown to be a prerequisite for intracellular replication, at least in some cell types (e.g., epithelial cells). Fundamental work has identified staphylococcal factors involved in these steps, such as fibronectin-binding proteins (FnBPs) for adhesion[15–17], the *agr* system and the downstream α-type phenol-soluble modulins (α-type PSMs) for vacuolar escape[18,19], hemolytic α-toxin for activation/subversion of the autophagic pathway[20], and hemolytic α-toxin and cysteine protease staphopain A for induction of intracellular cytotoxicity[10,21,22]. Nevertheless, a systematic investigation of the impact of most *S. aureus* factors on the bacterial intracellular lifestyle is still lacking.

Transposon mutant libraries are powerful tools for dissecting the genetic basis of diverse bacterial processes[23–25]. In the case of *S. aureus*, the Nebraska Transposon Mutant Library−NTML, has been a key resource for the identification of genes required for various functions, such as biofilm formation, resistance to antibiotics and antimicrobial peptides, host cell death, and bacterial growth in various conditions[26–33]. The NTML comprises 1920 individually mapped transposon-insertion mutants generated in the MRSA USA300 JE2 genetic background, covering virtually all the non-essential bacterial genes[27]. Notably, as an arrayed library, the NTML enables higher resolution phenotyping, including single-cell and time-resolved analyses, than the pooled libraries typically used for Tn-seq experiments.

Here, we profiled the infection of the NTML in epithelial cells using fluorescence microscopy-based infection assays and automated image analysis, at five times post-infection (from invasion up to 48 hpi), to identify bacterial factors that modulate *S. aureus* invasion, replication, persistence, and host cell death. We identified 73 *S. aureus* mutant strains with strong phenotypes, preserved across various non-professional phagocytes (osteoblasts, epithelial, and endothelial cells). Most of these factors have not been previously implicated in *S. aureus* intracellularity. Among these, we identified and characterized the *S. aureus* nicotinamidase PncA as a novel regulator of the *agr* system via modulation of the bacterial redox state, with a striking impact on virulence phenotypes. This comprehensive identification of *S. aureus* factors essential for its intracellular lifestyle constitutes a valuable resource for advancing studies on host-*S. aureus* interactions and holds significant potential for informing the development of targeted antimicrobial interventions against intracellular bacterial populations.

## Results

### Identification of bacterial factors regulating key features of the *S. aureus* intracellular lifestyle in epithelial cells

To systematically identify molecular determinants of *S. aureus* intracellular lifestyle upon infection of epithelial cells (HeLa229), we screened the Nebraska Transposon Mutant Library[27] (NTML; 1920 mutant strains arrayed in 384-well plates) by applying fluorescence microscopy-based infection assays and automated image analysis, as developed for our previous analysis of *S. aureus* clinical isolates[10] (Fig. 1a and Supplementary Fig. 1a). For each NTML mutant, we analysed the extent of infection, intracellular replication and host cell viability, at five times post-infection (0.5, 1.5, 3, 6 and 48 hpi), following *S. aureus* labelling with vancomycin BODIPY[34,35]; extracellular bacteria were eliminated by treatment with the bacteriolysin lysostaphin. The detailed time-course analysis of infection, replication, and host cell viability for all the *S. aureus* NTML mutant strains and WT is shown in Fig. 1b–d (strains showing strong phenotypes are highlighted in red/blue; WT strain USA300 JE2 is highlighted in black, for comparison); the entire screening dataset is provided in Supplementary Data 1). Two independent runs of the screening were performed, showing good reproducibility (Supplementary Fig. 2a and Supplementary Data 1).

Among the NTML mutants, only two strains presented reduced *S. aureus* invasion considering our stringent cutoff (≤1.5-fold infection at 0.5 hpi compared to WT), specifically NE353-*purH* and NE529-*purA* (Fig. 1b, highlighted in red). Following invasion, *S. aureus* has the ability to replicate within cells. We identified 58 mutants with reduced intracellular replication (≤3-fold maximum in replication compared to WT; Fig. 1c, highlighted in red), while 7 mutants showed enhanced replication (≥2-fold maximum in replication compared to WT; Fig. 1c, highlighted in blue). Moreover, the screening also identified *S. aureus* mutants with a strong impact on host cell viability, including 4 strains that induced faster kinetics of host cell death (≤2-fold host cell viability at 3 hpi compared to WT; Fig. 1d, highlighted in red), and 18 strains with reduced host cytotoxicity (≥2.5-fold minimum of host cell viability compared to WT; Fig. 1d, highlighted in blue). Finally, the extent of infection at 48 hpi was used as an indicator of *S. aureus* persistence, uncovering 12 mutants exhibiting high persistence (≥2-fold infection at 48 hpi and ≥2-fold minimum host cell viability compared to WT; Fig. 1b highlighted in blue). While 48 hpi represents an early timepoint for evaluating *S. aureus* persistence, it was the latest timepoint compatible/feasible for the large-scale microscopy-based infection assays in 384-well format. Representative microscopy images of cells infected with WT (USA300 JE2) and strains representative of different phenotypes are shown Fig. 1e.

Notably, some mutants were selected based on multiple phenotypes, e.g., showing simultaneously low intracellular replication, high intracellular persistence, and impaired host cell death. Collectively, a total of 101 mutant strains corresponding to 73 unique strains were selected for further characterization (Fig. 2a; selected strains highlighted in red/blue in Fig. 1b–d). While we recapitulated well known phenotypes of previously characterized mutants (cf. Discussion), most of the selected mutants have not been previously associated with *S. aureus* intracellularity.

The microscopy-based infection assays and automated image analysis protocols for the screening and follow-up experiments were optimized using WT strain and the two well-characterized mutants *agrA*[36–38] and *scpA*[10,22]. The *agrA* mutant, deficient in the *S. aureus* global virulence regulator *agr*, showed no replication within infected cells and exhibited high persistence and low host cytotoxicity, compared to WT (Supplementary Fig. 1b–d); the *scpA* mutant, deficient in the cysteine protease staphopain A, displayed very high and prolonged intracellular replication, and delayed onset of host cell death compared to WT (Supplementary Fig. 1b–d). These strains were included as internal controls in all assay plates to validate assay performance and to define thresholds for intracellular bacterial replication. In addition,

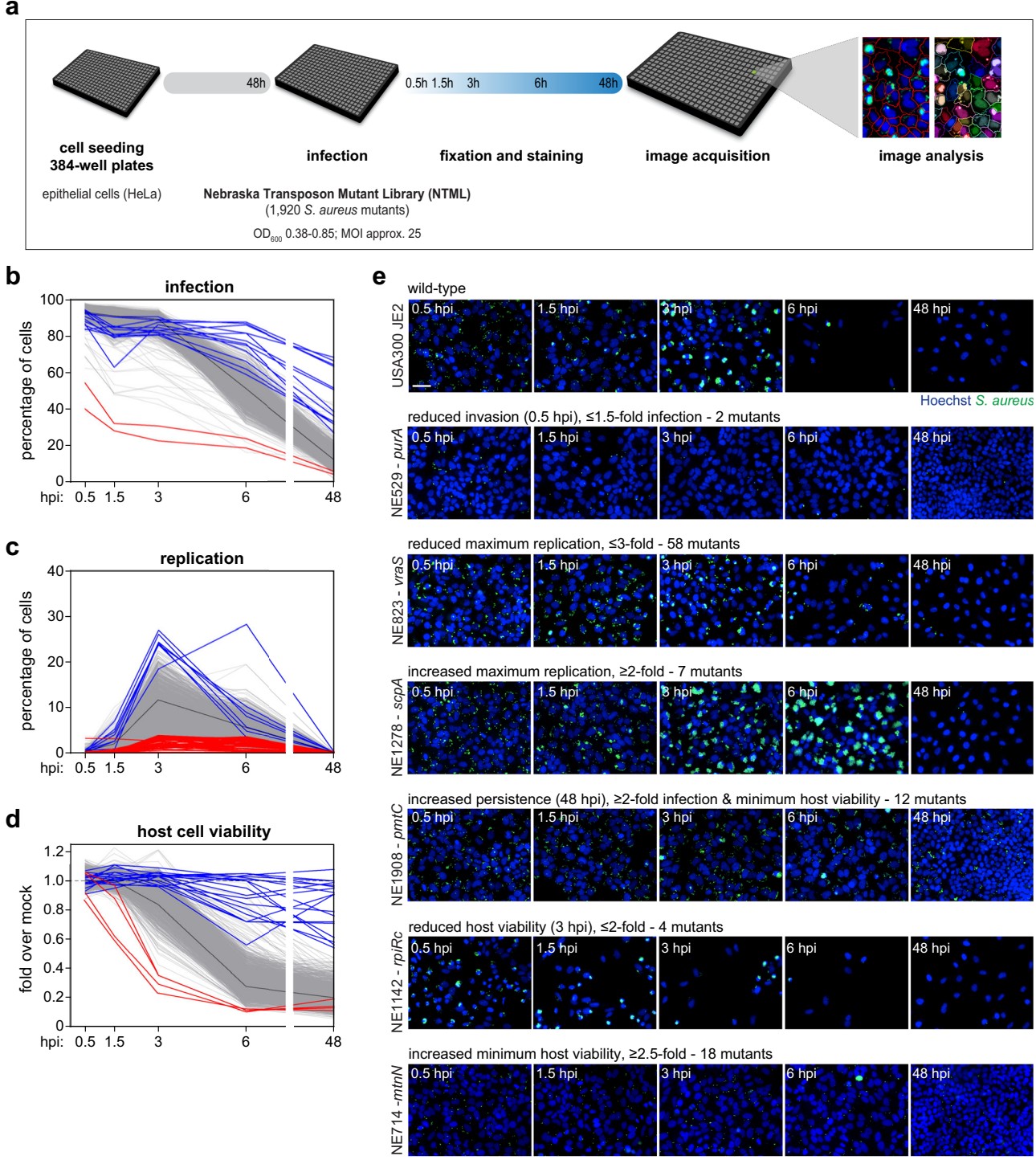

**Fig. 1 | High-throughput screening of the NTML identifies *S. aureus* factors relevant for the intracellular lifestyle within epithelial cells. a** Schematic representation of the workflow of the screening of the collection of 1920 *S. aureus* mutants (Nebraska transposon mutant library, NTML) to evaluate the impact of each bacterial factor on invasion, replication, persistence, and host cytotoxicity in epithelial cells (HeLa cells). Two independent runs of the screening were conducted; analysis was performed at 0.5, 1.5, 3, 6, and 48 h post-infection (hpi). **b**–**d** Time-course analysis of infection (**b**), intracellular replication (**c**), and host cell viability (**d**) of HeLa cells infected with all the individual *S. aureus* mutants. Cells infected with *S. aureus* WT (USA300 JE2) are shown for comparison (in black). Mutant strains with the strongest phenotypes are highlighted in blue (increase compared to WT) and red (reduction compared to WT); thresholds for strain selection are indicated in (**e**). For each strain, results are presented as the mean of two biologically independent experiments. **e** Representative fluorescence microscopy images of infection of HeLa cells with *S. aureus* WT and selected mutant strains exhibiting strong phenotypes, at 5 times post-infection (0.5, 1.5, 3, 6, and 48 hpi). Images are representative of two biologically independent experiments. Scale bar, 50 μm.

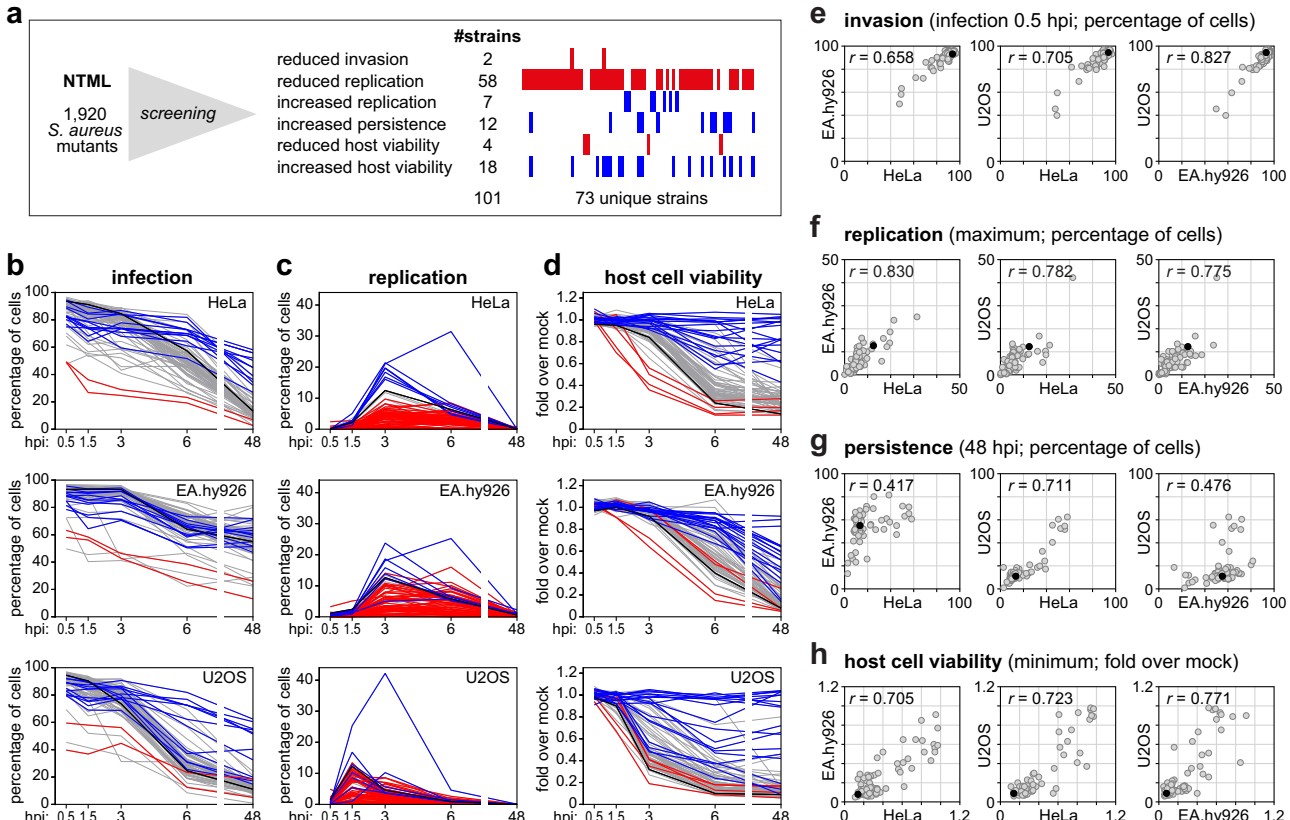

**Fig. 2 | *S. aureus* mutants exhibit similar intracellular lifestyles in various non-phagocytic cells. a** Schematic representation of the overlap of the intracellular phenotypes of the 101 *S. aureus* strains selected based on the thresholds indicated in Fig. 1e, leading to the selection of 73 unique strains for further characterization. **b**–**d** Time-course analysis of infection (**b**), intracellular replication (**c**), and host cell viability (**d**) for the 73 selected *S. aureus* mutant strains in epithelial cells (HeLa), endothelial cells (EA.hy926), and osteoblasts (U2OS). Strains highlighted in blue and red correspond to the strains selected based on the screening results in HeLa cells, highlighted in Fig. 1b–d. Cells infected with *S. aureus* WT are shown for comparison (black line). For each strain, results are presented as the mean of three biologically independent experiments. **e**–**h** Pairwise comparison of the percentage of invasion (infection at 0.5 hpi; **e**), cells with high *S. aureus* intracellular replication (maximum value; **f**), persistence (infection at 48 hpi; **g**), and host cell viability (minimum value; **h**) upon infection of the three cell types. Cells infected with *S. aureus* WT are shown for comparison (black dot). Results are presented as the mean of three biologically independent experiments. Spearman's rank correlation coefficients are shown in the upper left corner of each graph.

the eFluor proliferation assay, based on dilution of the fluorescence dye with bacterial replication[39], was used to assist in defining the thresholds for intracellular replication (Supplementary Fig. 1e). Prior to infection, *S. aureus* mutants were grown to exponential phase ($OD_{600}$ 0.38–0.85, median 0.51), a range confirmed not to influence infection assay parameters (Supplementary Fig. 2b–d).

Overall, through the screening of the NTML, we identified multiple *S. aureus* factors crucial for distinct features of the intracellular lifestyle, the majority of which were not previously associated with this phenotype.

### *S. aureus* factors modulate bacterial intracellular lifestyle across various non-phagocytic cells

The kinetics and levels of intracellular replication, persistence, and host cell death upon infection with *S. aureus* vary depending on the host cell type[10,40]. While the most pronounced differences occur between professional and non-professional phagocytes, relevant discrepancies have also been observed among different non-professional phagocytic cell types. For instance, the persistence of *S. aureus* is higher in endothelial cells compared to osteoblasts and epithelial cells, and the ability of *S. aureus* clinical isolates to induce host cell death is highly variable across epithelial, endothelial, and osteoblast cells.

To investigate whether the phenotypes associated with the 73 selected *S. aureus* mutants are broadly relevant across non-

professional phagocytic cells, we conducted a comparative analysis of their intracellular phenotypes (infection, replication, host cell viability; Fig. 2b–d and Supplementary Data 2) in epithelial cells (HeLa229), endothelial cells (EA.hy926), and osteoblasts (U2OS). Pairwise comparisons of invasion (infection at 0.5 hpi), maximum intracellular replication, and minimum host cell viability revealed strong correlations across the cell types analysed (Spearman's $r > 0.65$; Fig. 2e, f, h). Persistence (assessed as infection at 48 hpi) was also strongly correlated between epithelial cells and osteoblasts (Spearman's $r = 0.711$; Fig. 2g), though lower correlations were observed between these and endothelial cells (Spearman's $r < 0.48$; Fig. 2g). For epithelial cells, the results presented were obtained from follow-up/validation experiments conducted independently of the original screen, showing very strong correlation with the screening dataset (Supplementary Fig. 3a–d).

Of note, we assessed the bacterial fluorescence integrated intensity at 5 minutes post-infection (used as a proxy for the initial bacterial intracellular load), which showed a low correlation with the maximum intracellular replication observed for each mutant in epithelial cells (Supplementary Fig. 3e). This finding indicates that variations in bacterial invasion efficiency do not account for the differences in replication across the 73 selected *S. aureus* mutants. Additionally, no substantial differences in growth in liquid medium (TSB) were observed among the 73 selected *S. aureus* mutants (Supplementary Fig. 3f).

To address potential limitations associated with the use of transformed cell lines, we also profiled the infection of the 73 selected *S. aureus* mutant strains in primary human umbilical vein endothelial cells (HUVEC; Supplementary Fig. 4a–c and Supplementary Data 2). Overall, the intracellular phenotypes observed in HUVECs were comparable to those in transformed cell lines (infection, replication, and host cell viability with Spearman's *r* > 0.64; Supplementary Fig. 4d, e, g). Nonetheless, persistence in HUVECs correlated poorly with that observed in all transformed cell lines tested (Spearman's *r* < 0.37; Supplementary Fig. 4f).

Taken together, these results demonstrate that the 73 selected *S. aureus* factors play consistent roles in modulating the bacterial intracellular lifestyle in non-professional phagocytic cells. Notwithstanding, the observed differences, particularly in persistence, underscore the influence of host cell type on the functional relevance of these bacterial determinants.

### Intracellular phenotypic profiles stratify *S. aureus* mutants into distinct clusters

Hierarchical cluster analysis based on the phenotypic analysis of the 73 selected *S. aureus* mutant strains in epithelial, endothelial and osteoblast cells, at the 5 timepoints analyzed, revealed 8 groups with distinct profiles (Fig. 3a). The main phenotypic characteristics of these clusters in the three cell types tested are summarized in Fig. 3b–d (median profiles), and Fig. 3e–g and Supplementary Fig. 5a, b (key phenotypes); representative images of mutant strains belonging to the various clusters are shown in Supplementary Fig. 6. Comparable infection phenotypes for each cluster were obtained in HUVEC (Supplementary Fig. 4h–k).

While strains belonging to cluster a (marked in orange) showed reduced invasion across all cell types compared to WT strain, mutants belonging to all other clusters efficiently invaded host cells, but displayed varying levels of intracellular replication, persistence, and ability to induce host cell death (Fig. 3 and Supplementary Fig. 5a). For instance: i) mutant strains from clusters b and h (marked in lime and green, respectively) exhibited reduced intracellular replication and reduced host cytotoxicity, though only strains belonging to cluster h presented high persistence; or ii) while mutant strains from clusters b, c, d, f and g (marked in lime, pink, cyan, red and blue, respectively) showed low intracellular replication, mutants from cluster c (marked in pink) were distinguished by faster kinetics of host cell death (Fig. 3 and Supplementary Fig. 5b). Two mutant strains were not included in any of the identified clusters (Fig. 3), specifically: strain NE828-*aspB* (disrupted in gene *aspB*−SAUSA300_RS10490, encoding for a hypothetical protein of unknown function), which presented low replication and high persistence (comparable to cluster h, marked in green), but strong reduction of host cell viability (comparable to WT strain); and strain NE1278-*scpA* (mutant for *scpA*−SAUSA300_RS10340, encoding the cysteine protease staphopain A) that presented the highest levels intracellular replication and a delayed onset of host cell death, when compared with the strains included in cluster e (marked in violet), the group with the highest replication.

We anticipate that the leveraging of existing knowledge of the characterized factors within the different phenotypic clusters will offer insights into the functions and mechanisms of action of uncharacterized neighboring mutants. An illustrative example is cluster c (marked in pink), which includes mutants in *rot* and *rpiRc*, well-characterized repressors of virulence factor expression (e.g., leukocidins, secreted proteases)[41–45]. Infection with mutants of these repressors has previously been shown to result in increased cell death of neutrophils and reduced survival in mouse models[42,44–46]. Interestingly, cluster c also includes a mutant in *glcT*, encoding a predicted transcriptional antiterminator[47,48]. This gene has not been previously characterized in *S. aureus* or associated to regulation of host cell death. Another compelling example is cluster h (marked in green), which includes seven

mutants previously associated with the *agr* system and α-type PSMs secretion (*agrA*, *agrB*, *agrC*, *pmtB*, *pmtC*, *abcA*, *clpP*), alongside a mutant in *pncA*, which has not been characterized previously (see below).

The main features of the different clusters and strains were validated using alternative methods in epithelial cells, namely colony-forming unit assays for the intracellular bacterial load (for at least three mutant strains per group; Fig. 3h) and luminogenic ATP assay to assess host cell viability (for the 73 mutants; Supplementary Fig. 5c,d). The results of these validation experiments were consistent with the findings from the microscopy-based infection assays.

Overall, the use of multiple cell types provided a more granular phenotypic profiling of the 73 selected *S. aureus* mutants, revealing interesting clusters of mutants with distinct intracellular profiles upon infection of non-professional phagocytes. Moreover, the strains (with known functions) gathered within each cluster provide important insights into their possible function and/or mechanisms of action.

### *S. aureus* mutants belonging to cluster h present low vacuolar escape and are associated with *agr* or α-type PSM export deficiency

Intracellular bacterial pathogens often employ strategies to evade the degradative environment of host cell vacuoles, seeking the nutrient-rich cytoplasmic environment for replication and survival[49]. *S. aureus* has also been reported to exhibit this behavior in certain cell types, including epithelial and endothelial cells[18,19,50,51]. Indeed, an association between vacuolar escape and induction of host cell death in non-professional phagocytic cells has been proposed[10,38,40,50]. Considering this, we assessed the vacuolar vs. cytosolic localization of the 73 selected mutant strains to determine how bacterial subcellular localization within the host cell is related to the strong phenotypes described above. For this purpose, we evaluated the recruitment of fluorescently labelled C-terminal cell wall-targeting (CWT) domain of lysostaphin to *S. aureus* (Fig. 4a, b Supplementary Fig. 7a, c and Supplementary Data 3), which is indicative of cytosolic localization of the bacteria[18,51], using cells stably expressing mRFP-CWT. In addition, we evaluated the colocalization between *S. aureus* and LysoTracker (Fig. 4c, d Supplementary Fig. 7b, d and Supplementary Data 3), a widely used marker of endo-lysosomes/acidic vesicles. These analyses were performed at the time post-infection that precedes the peak of intracellular replication, specifically 1.5 hpi for epithelial (HeLa) and endothelial (EA.hy926) cells and 0.5 hpi for osteoblasts (U2OS). As expected, the extent of *S. aureus*/CWT and *S. aureus*/LysoTracker colocalizations showed an inverse correlation (Supplementary Fig. 7e–g).

The vast majority of the selected *S. aureus* mutant strains showed a translocation to the cytosol similar to WT, independently of the cell type (Fig. 4a, c and Supplementary Fig. 7a–d). This indicates that the reduced intracellular replication exhibited, for example, by strains belonging to clusters b and g (marked in lime and blue, respectively) is not related to their inability to reach the host cytosol. Along this line, we observed low/moderate correlations between cytosolic localization and intracellular replication (Supplementary Fig. 7h, i). Higher correlations were observed between vacuolar escape and host cell death (Supplementary Fig. 7j, k), as previously reported[10,38,40,50].

Notably, strains from cluster h (marked in green) and NE828-*aspB* showed the lowest levels of *S. aureus*/CWT colocalization and the highest *S. aureus*/LysoTracker colocalization, indicating the predominant localization of these strains inside vacuoles (Fig. 4a, c and Supplementary Fig. 7a–d). Consistent with the impaired ability to escape the vacuole, the *S. aureus* mutants from cluster h exhibited low intracellular replication, high persistence, and low host cell death upon infection of epithelial cells, endothelial cells, and osteoblasts (Figs. 3 and 4f–h and Supplementary Fig. 8a–c). These mutants also presented low hemolytic activity (Supplementary Fig. 8d, e). Cluster h

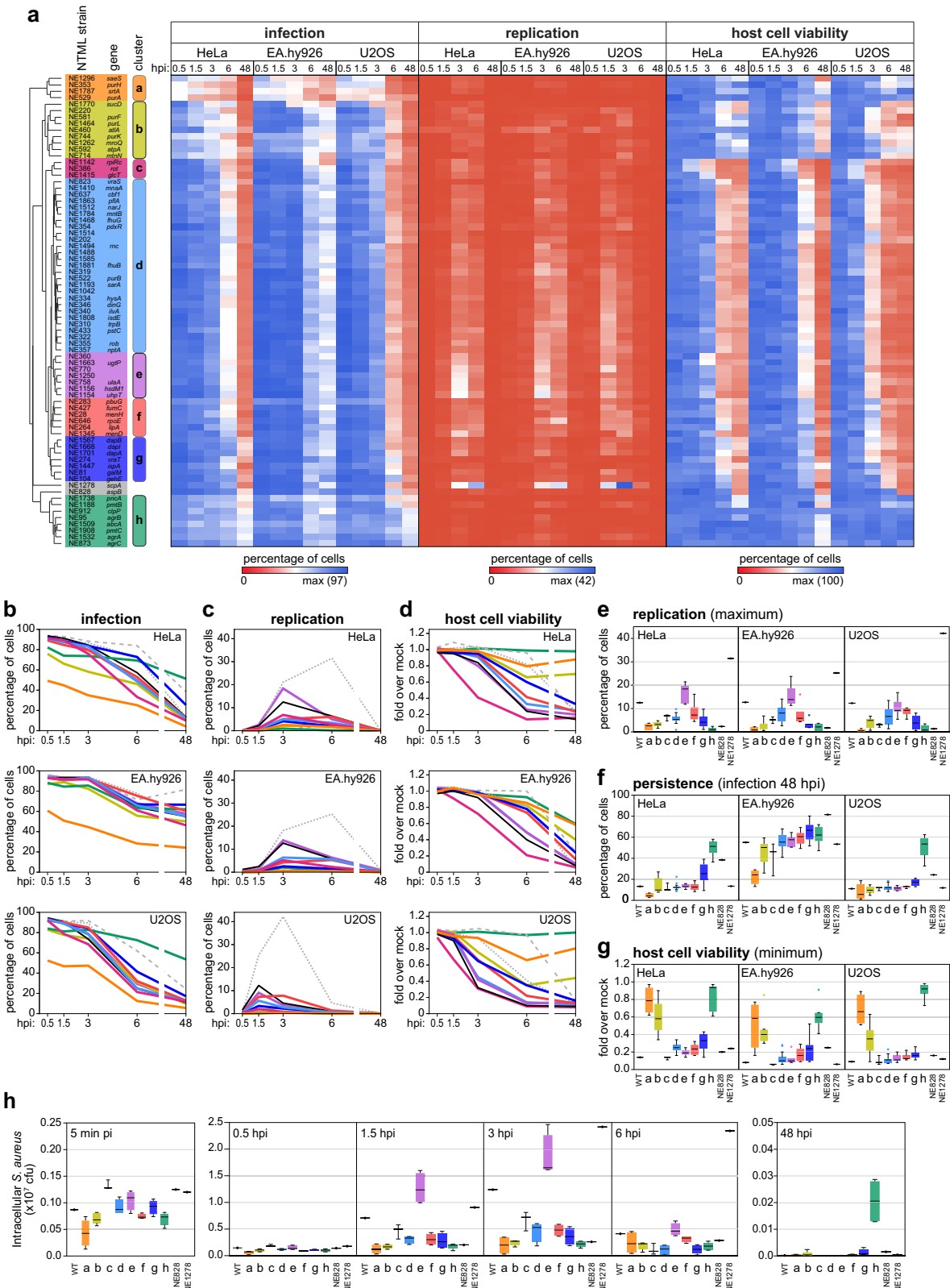

includes a mutant in *pncA* (previously uncharacterized function in *S. aureus*, see below), as well as mutants of the quorum sensing system genes *agrA*, *agrB*, and *agrC*, the major protease and *agr* system regulator gene *clpP*[52], and the PSM transporter genes *pmtB*, *pmtC*, and *abcA*[53,54] (Fig. 4e). α-type PSMs are crucial players in *S. aureus* vacuolar

escape, and their expression is transcriptionally regulated by the *agr* system[18,55].

Together, these results indicate that mutations leading to *agr* or α-type PSM export deficiency are the predominant determinants for reduced *S. aureus* vacuolar escape. Moreover, our data demonstrates

**Fig. 3 | Selected *S. aureus* factors regulate multiple aspects of the intracellular lifestyle. a** Heat map and hierarchical clustering of the phenotypic profiles (infection, intracellular replication, and host cell viability) of the 73 selected *S. aureus* mutant strains upon infection of epithelial cells (HeLa), endothelial cells (EA.hy926), and osteoblasts (U2OS), at 5 times post-infection (0.5, 1.5, 3, 6, and 48 hpi). Results are presented as the mean of three biologically independent experiments. Hierarchical clustering of the phenotypic profiles exhibited by the bacterial strains was performed based on Euclidean distance. Eight clusters with distinct infection profiles were identified, labelled a-h and highlighted in different colours; two mutant strains were not included in any clusters, specifically strains NE828-*aspB* and NE1278-*scpA*. NTML strain ID and the corresponding disrupted gene symbol are indicated; gene locus tag is indicated in Supplementary Data 1–3. **b–d** Time-course analysis of infection (**b**), intracellular replication (**c**), and host cell viability (**d**) in epithelial cells (HeLa), endothelial cells (EA.hy926) and osteoblasts (U2OS) for each phenotypic cluster identified in panel a. For each cluster, results are presented as the median of all the strains belonging to the respective cluster for each timepoint, generated using the mean of three biologically independent experiments per strain. Results are coloured by phenotypic group (as defined in Fig. 3a); WT (black line), NE828-*aspB* (grey dashed line), and NE1278-*scpA* (grey

dotted line) are also shown. **e–g** Box-plots showing the distribution of data for *S. aureus* WT and 73 selected mutant strains concerning intracellular replication (maximum value; **e**), persistence (infection at 48 hpi; **f**), and host cell viability (minimum; **g**) upon infection of the three non-professional phagocytic cells. Box-plots were generated using the mean of three biologically independent experiments per strain and are stratified by phenotypic group; white lines show the medians, box limits indicate the 25th–75th percentiles, whiskers extend 1.5 times the interquartile range from the 25th to 75th percentiles. **h** Box-plots showing the distribution of data for quantification of intracellular *S. aureus* by CFU assays upon infection of HeLa cells with a subset of bacterial strains belonging to each phenotypic cluster (3 strains from cluster c, 4 strains from clusters a, d, e, f, g, 5 strains from cluster b, and 6 strains from cluster h; strains NE828 and NE1278 are also included; total 36 mutant strains). CFUs for cells infected with *S. aureus* WT are shown for comparison. Infection was analysed at the indicated times (0.5, 1.5, 3, 6, and 48 hpi); box-plots were generated using the mean of three biologically independent experiments per strain, and are coloured by phenotypic group; white lines show the medians, box limits indicate the 25th–75th percentiles, whiskers extend 1.5 times the interquartile range from the 25th to 75th percentiles. Source data are provided as a Source data file.

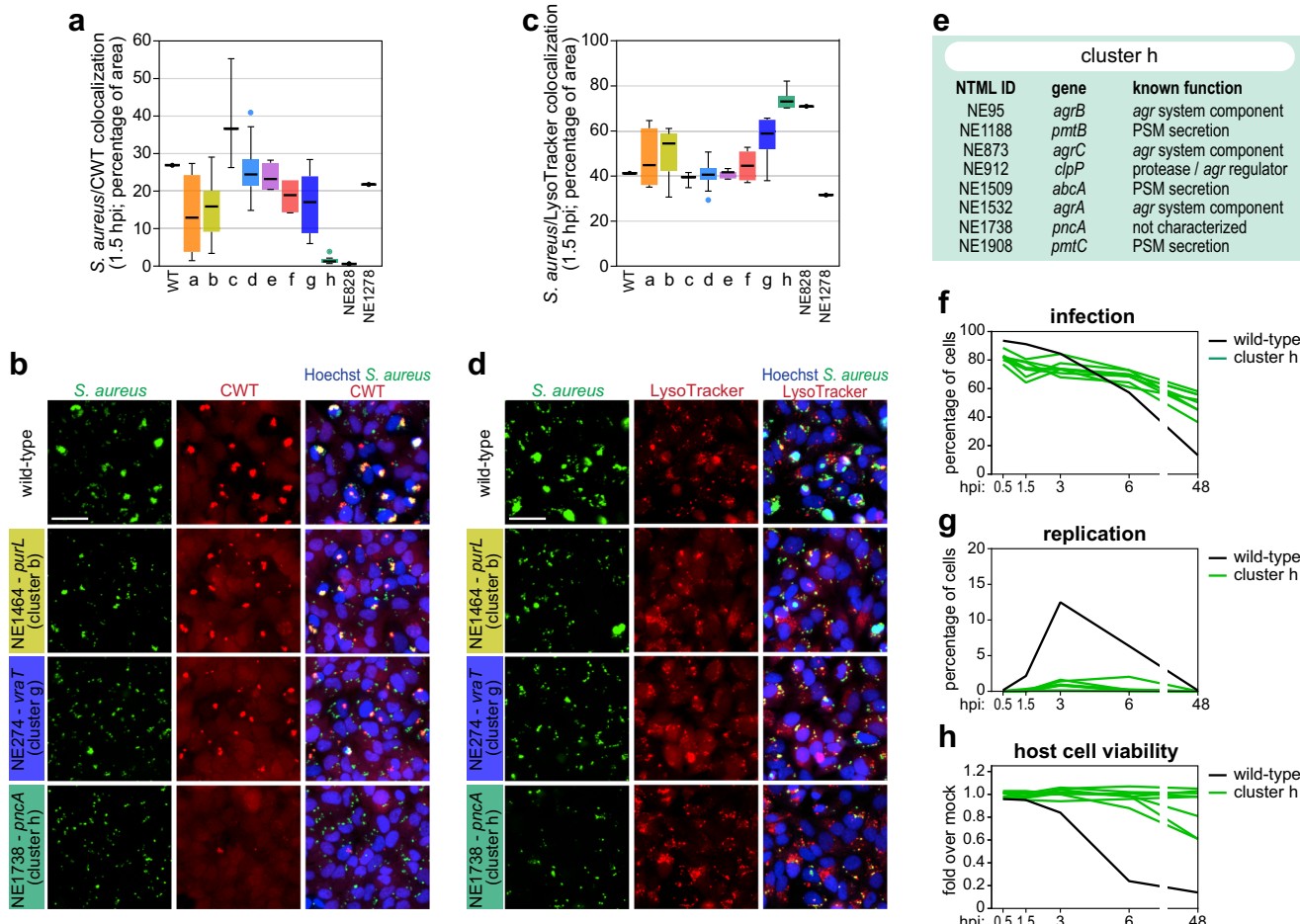

**Fig. 4 | A cluster of *S. aureus* mutants showing low vacuolar escape is associated with impairment of *agr* activity or α-type PSM export. a, c** Box-plots showing the distribution of data for *S. aureus* WT and 73 selected mutant strains concerning *S. aureus*/CWT (**a**) and *S. aureus*/LysoTracker (**c**) colocalization in HeLa cells analyzed at 1.5 hpi. Results are coloured by phenotypic group; box-plots were generated using the mean of three biologically independent experiments per strain; white lines show the medians, box limits indicate the 25th–75th percentiles, whiskers extend 1.5 times the interquartile range from the 25th and 75th percentiles. **b, d** Representative fluorescence microscopy images of HeLa cells infected with *S. aureus* WT and strains belonging to cluster b (NE1464-*purL*), g (NE274-*vraT*), and h

(NE1738-*pncA*), to evaluate the recruitment of fluorescently-labelled CWT (**b**), and colocalization with LysoTracker (**d**). Microscopy images are representative of three biologically independent experiments and correspond to 1.5 hpi. Scale bar, 50 μm. **e** Table indicating the 8 *S. aureus* strains belonging to cluster h, including NTML strain ID, disrupted gene, and known function. **f–h** Time-course analysis of infection (**f**), intracellular replication (**g**), and host cell viability (**h**) upon infection of HeLa cells with the 8 *S. aureus* mutant strains belonging to cluster h (green lines). Cells infected with *S. aureus* WT are shown for comparison (black line). For each strain, results are presented as the mean of three biologically independent experiments. Source data are provided as a Source data file.

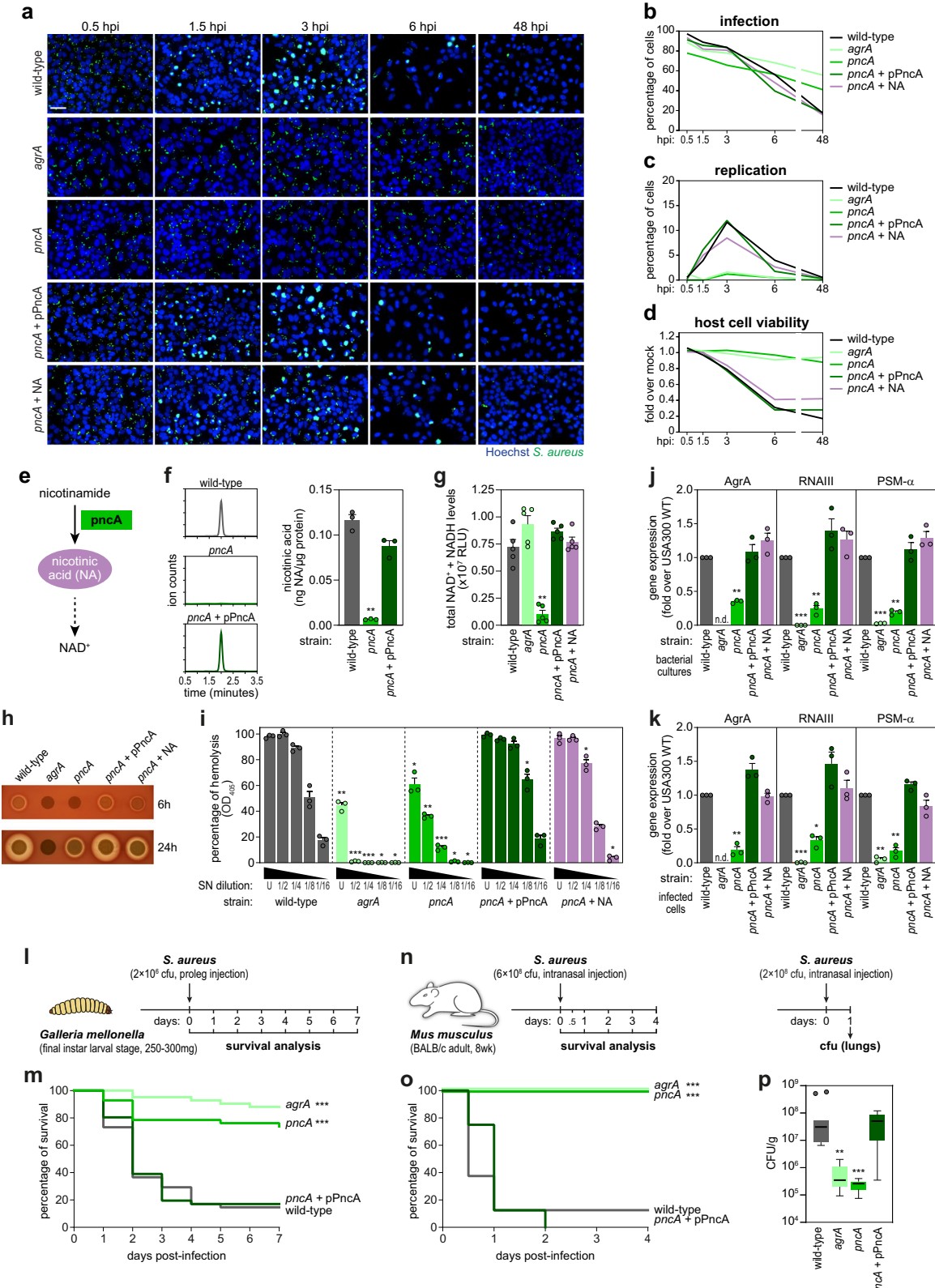

that cytosolic access per se is insufficient to enact intracellular replication.

### The *S. aureus* nicotinamidase PncA is a novel regulator of *agr* activity and virulence, via redox state modulation

As a case study, we focused on the uncharacterized mutant present within cluster h−NE1738-*pncA*. This mutant displayed low vacuolar

escape, low intracellular replication, high persistence, and low host cytotoxicity in non-professional phagocytic cells, which were restored to a WT-like profile in the complemented strain (Fig. 5a–d and Supplementary Fig. 9a–h). The *pncA* mutant also exhibited pronounced phenotypes in macrophages (differentiated THP1 cells), including strongly reduced intracellular replication, and increased persistence and host cell viability (Supplementary Fig. 9i–l).

**Fig. 5 | The *S. aureus* nicotinamidase PncA regulates *agr* system activity.**
**a–d** Representative fluorescence microscopy images (**a**) and time-course analysis of infection (**b**), intracellular replication (**c**), and host cell viability (**d**) upon infection of HeLa cells with *S. aureus* WT, mutant strains *agrA*, *pncA*, *pncA*+pPncA, and *pncA* in medium supplemented with nicotinic acid (*pncA* + NA). Results shown in **b**–**d** are the mean of three biologically independent experiments; microscopy images are representative of three biologically independent experiments. Scale bar, 50 μm. **e** Schematic representation of the PncA activity as a nicotinamidase. **f** Chromatogram and quantification by liquid chromatography-mass spectrometry of nicotinic acid in stationary phase cultures of *S. aureus* WT, *pncA*, and *pncA* +pPncA. Results are presented as the mean ± s.e.m. of three biologically independent experiments; **$P < 0.01$ (statistical analysis is detailed in Supplementary Data 4). **g** Quantification of total NAD$^+$ & NADH levels in stationary phase cultures of *S. aureus* WT, *agrA*, *pncA*, *pncA*+pPncA, and *pncA* + NA. Results are presented as the mean ± s.e.m. of five biologically independent experiments; **$P < 0.01$ (statistical analysis is detailed in Supplementary Data 4). **h, i** Haemolytic activity of *S. aureus* WT, *agrA*, *pncA*, *pncA*+pPncA, *pncA* + NA upon spotting of liquid cultures onto TSB agar containing 2% sheep blood (**h**) or quantification of OD$_{405}$ of a 2% sheep blood solution upon incubation with the strains supernatants (**i**). Culture supernatants were used undiluted (U) or after 2-fold serial dilutions (up to 1/16). All values are shown normalized to the OD$_{405}$ of blood incubated with TSB containing 1% Triton X-100, and presented as mean ± s.e.m. of three biologically independent experiments; *$P < 0.05$, **$P < 0.01$, and ***$P < 0.001$ (statistical analysis is detailed in Supplementary Data 4). **j, k** Expression levels of AgrA, RNAIII, and PSM-α determined by qRT-PCR in liquid cultures of *S. aureus* WT, *agrA*, *pncA*, *pncA*+pPncA, and *pncA* + NA (**j**), or upon infection of HeLa cells with the same strains and collected at 3 hpi (**k**). Results are shown normalized to *S. aureus* WT, and presented as mean ± s.e.m. of three biologically independent experiments; *$P < 0.05$, **$P < 0.01$, and ***$P < 0.001$ (statistical analysis is detailed in Supplementary Data 4). **l, n** Schematic representation of the workflow used for the *Galleria mellonella* larvae (**l**) and mouse pneumonia (**n**) *S. aureus* infection models. **m** Seven-day survival curves of *Galleria mellonella* following infection with *S. aureus* WT, *agrA*, *pncA*, and *pncA*+pPncA. Results are presented as the mean of four biologically independent experiments performed with 10 larvae per strain. ***$P < 0.001$ (statistical analysis is detailed in Supplementary Data 4). **o, p** Survival curves (**o**) and box-plot showing the distribution of the CFU data from lungs (**p**) of mice infected with *S. aureus* WT, *agrA*, *pncA*, and *pncA*+pPncA via nasal instillation ($n = 8$ per group for survival and $n = 11$ per group for CFUs). Lungs were collected 24 h after infection. **$P < 0.01$, and ***$P < 0.001$ (statistical analysis is detailed in Supplementary Data 4). Source data are provided as a Source data file.

PncA is a putative nicotinamidase involved in the production of NAD$^+$, specifically catalysing the conversion of nicotinamide to nicotinic acid (NA; Fig. 5e), as inferred from homology with other bacteria[56,57]. Accordingly, we observed that NA and NAD$^+$/NADH levels were significantly reduced in the *pncA* mutant and restored to WT levels upon complementation (Fig. 5f, g). Additionally, the *pncA* mutant failed to grow in NA-deficient media (RPMI), but supplementation with NA restored the growth of the mutant to the levels achieved by the WT and complemented strains (Supplementary Fig. 9m). In contrast, only a mild growth defect was observed in TSB, due to its NA content, which bypasses the requirement for PncA in NAD$^+$ biosynthesis until the NA present in the medium is depleted (Supplementary Fig. 9n). Nicotinic acid supplementation also restored the intracellular infection profile and NAD$^+$/NADH levels of the *pncA* mutant to that of the WT strain (Fig. 5a–d, g). Altogether, these results confirm PncA as a bona fide nicotinamidase and confirm the relevance of this factor to intracellular *S. aureus*.

The phenotypic similarity and integration of the *pncA* mutant in cluster h alongside *agr*-related factors prompted us to investigate how PncA intersects the *agr* system. In line with this, the *pncA* mutant exhibited reduced haemolytic activity compared to WT and complemented strains, resemblant of the *agrA* mutant (Fig. 5h, i). Furthermore, the *pncA* mutant showed lower expression of *agr* core genes and effectors (AgrA and RNAIII, respectively), as well as of downstream genes (PSM-α), when compared to WT, both in bacterial cultures (Fig. 5j) and infected cells (Fig. 5k).

To determine if the pronounced in vitro phenotypes of the *pncA* mutant have implications for *S. aureus* infection in vivo, we employed two *S. aureus* infection models: *Galleria mellonella* and mouse pneumonia (Fig. 5l, n). Notably, the *pncA* mutant displayed significantly reduced virulence compared to the WT and complemented strains in both models (Fig. 5m, o, p). In fact, survival of larvae and mice infected with the *pncA* mutant was comparable to that observed for the *agrA* mutant (Fig. 5m, o, p), previously shown to have attenuated virulence[58–61]. These findings are in complete agreement with the in vitro data showing reduced intracellular replication and host cytotoxicity induced by infection with the *pncA* (and *agrA*) mutants, when compared to the WT strain.

Given the role of NAD$^+$/NADH ratio in redox homeostasis[56,57] and the known inactivation of the *agr* system under oxidizing conditions[62], we hypothesized that the *pncA* mutant presents an unbalanced redox state leading to *agr* inhibition. Indeed, the *pncA* mutant showed significantly elevated ROS levels compared to WT and complemented strains (Fig. 6a), as evaluated using the chemical probe CM-H2DCFDA.

Treatment with the antioxidant N-acetyl-L-cysteine (NAC) reduced ROS levels to those of the WT strain (Fig. 6a). Importantly, NAC treatment restored the hemolytic activity (Fig. 6b, c), and intracellular profile of the *pncA* mutant to those of the WT and complemented strains (Fig. 6d–h). In agreement, NAC treatment of the *pncA* mutant reactivated *agr*, as evidenced by the restoration of AgrA, RNAIII, and PSM-α expression to WT levels (Fig. 6i bacterial cultures, Fig. 6j infected cells). Along this line, the levels of BsaA, a glutathione peroxidase essential for bacterial resistance to oxidative stress and normally transcriptionally repressed by AgrA in non-oxidative conditions[62], were elevated in the *pncA* mutant and returned to WT levels upon NAC treatment (Fig. 6i, j). Of note, a *pncA/agr* mutant showed reduced intracellular replication, increased persistence, and high host cell viability upon infection of epithelial cells (Supplementary Fig. 10a–d), resembling the profiles observed for the single mutants (Fig. 5a–d). Consistent with the absence of the *agr* operon, NAC treatment had no effect on the *pncA/agr* mutant (Supplementary Fig. 10a–d). These results indicate that the *pncA* mutant phenotypes are caused by increased ROS triggered by NAD$^+$/NADH imbalance, rather than by a lack of NAD$^+$ as co-factor in metabolic pathways.

To further evaluate the relevance of PncA in the intracellular lifestyle of *S. aureus*, we generated *pncA* mutants in clinical isolates from bacteraemia (Ba), bone/joint infections (BJI), and infective endocarditis (IE), previously characterized for their intracellular profiles[10]. As expected, across all backgrounds, the *pncA* mutants showed lower NA and NAD$^+$/NADH levels, no growth in RPMI, reduced haemolytic activity, and diminished expression of *agr* core and downstream genes in bacterial cultures (Supplementary Fig. 10e–j). The *pncA* mutants exhibited low intracellular replication, high persistence, and reduced host cell death during epithelial cell infection (Fig. 7a–d), which were restored to intracellular profiles comparable to with their corresponding parental isolates upon NAC treatment (Fig. 7a–d).

Collectively, these findings demonstrate that the nicotinamidase PncA modulates the intracellular lifestyle and virulence of *S. aureus* by regulating *agr* system activity through its impact on the bacterial redox state (Fig. 7e).

## Discussion

Despite the increasing recognition of *S. aureus* as a facultative intracellular pathogen, our understanding of the bacterial factors underpinning this lifestyle remains incomplete. To address this, we conducted a comprehensive and systematic, multiparametric analysis of the intracellular lifestyle of a well-defined arrayed collection of *S.*

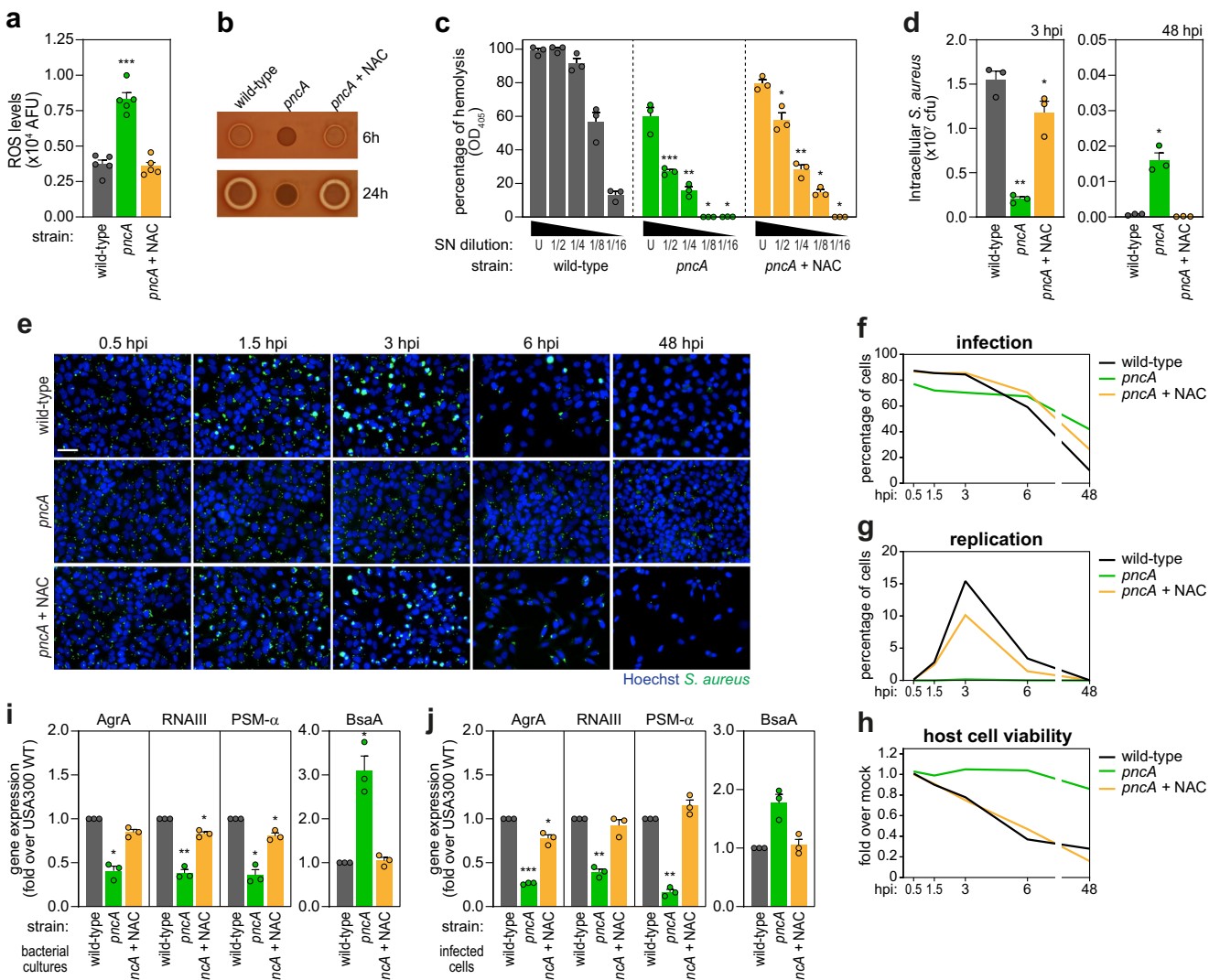

**Fig. 6 | PncA modulates *S. aureus* virulence through regulation of the bacterial redox state. a** Quantification of ROS in exponential phase cultures of *S. aureus* WT, *pncA*, and *pncA* in medium supplemented with NAC (*pncA* + NAC). Results are presented as the mean ± s.e.m. of five biologically independent experiments; ***P < 0.001 (statistical analysis is detailed in Supplementary Data 4). **b, c** Hemolytic activity of *S. aureus* WT, *pncA*, and *pncA* + NAC upon spotting of liquid cultures onto TSB agar containing 2% sheep blood (**b**) or quantification of OD$_{405}$ of a 2% sheep blood solution upon incubation with the strains supernatants (**c**). Culture supernatants were used undiluted (U) or after 2-fold serial dilutions (up to 1/16). All values are shown normalized to the OD$_{405}$ of blood incubated with TSB containing 1% Triton X-100, and presented as mean ± s.e.m. of three biologically independent experiments; *P < 0.05, **P < 0.01, and ***P < 0.001 (statistical analysis is detailed in Supplementary Data 4). **d** Quantification of intracellular *S. aureus* by CFU assays upon infection of HeLa cells with *S. aureus* WT, *pncA*, and *pncA* + NAC, and analysed at 3 and 48 hpi. Results are presented as the mean ± s.e.m. of three biologically independent experiments; *P < 0.05 and **P < 0.01 (statistical analysis is detailed in Supplementary Data 4). **e–h** Representative fluorescence microscopy images (**e**) and time-course analysis of infection (**f**), intracellular replication (**g**), and host cell viability (**h**) upon infection of HeLa cells with *S. aureus* WT, *pncA*, and *pncA* + NAC. Results shown in **f–h** are the mean of three biologically independent experiments; microscopy images are representative of three biologically independent experiments. Scale bar, 50 μm. **i, j** Expression levels of AgrA, RNAIII, PSM-α, and BsaA were determined by qRT-PCR in liquid cultures of *S. aureus* WT, *pncA*, and *pncA* + NAC (**i**), or upon infection of HeLa cells with the same strains and collected at 3 hpi (**j**). Results are shown normalized to *S. aureus* WT, and presented as mean ± s.e.m. of three biologically independent experiments; *P < 0.05, **P < 0.01, and ***P < 0.001 (statistical analysis is detailed in Supplementary Data 4). Source data are provided as a Source data file.

*aureus* mutants (NTML, 1920 mutant strains) upon infection of epithelial cells, with high temporal resolution, at single-cell level. This analysis was performed at five times post-infection (from invasion to 48 hpi) and enabled the identification of bacterial mutants with defects in key intracellular phenotypes, specifically invasion, intracellular replication, persistence, and ability to induce host cell death. Notably, 307 of the 1920 mutants exhibited alterations in at least one of the phenotypic characteristics evaluated (ca. 16%, considering a 2-fold threshold relative to control).

Further experiments performed in non-professional phagocytes (osteoblasts, epithelial and endothelial cells) using a subset of 73 *S. aureus* mutants that scored among the strongest hits in the screening revealed that their intracellular phenotypes were largely conserved, particularly for invasion, intracellular replication, and host cell viability. These findings are in line with previous reports and our own observations using *S. aureus* clinical isolates[10]. Nevertheless, given that the selection of mutants was based on their strong phenotypes in epithelial cells, it is plausible that other factors may exhibit host cell type-specific roles. Indeed, the observed variations in persistence across osteoblasts, epithelial, and endothelial cells underscore the importance of the host cell type in shaping *S. aureus* intracellular phenotypes. Of note, the 48 hpi timepoint used in this study is limiting for assessing persistence, and future studies will require longer-term experiments and in vivo models optimized for chronic infection.

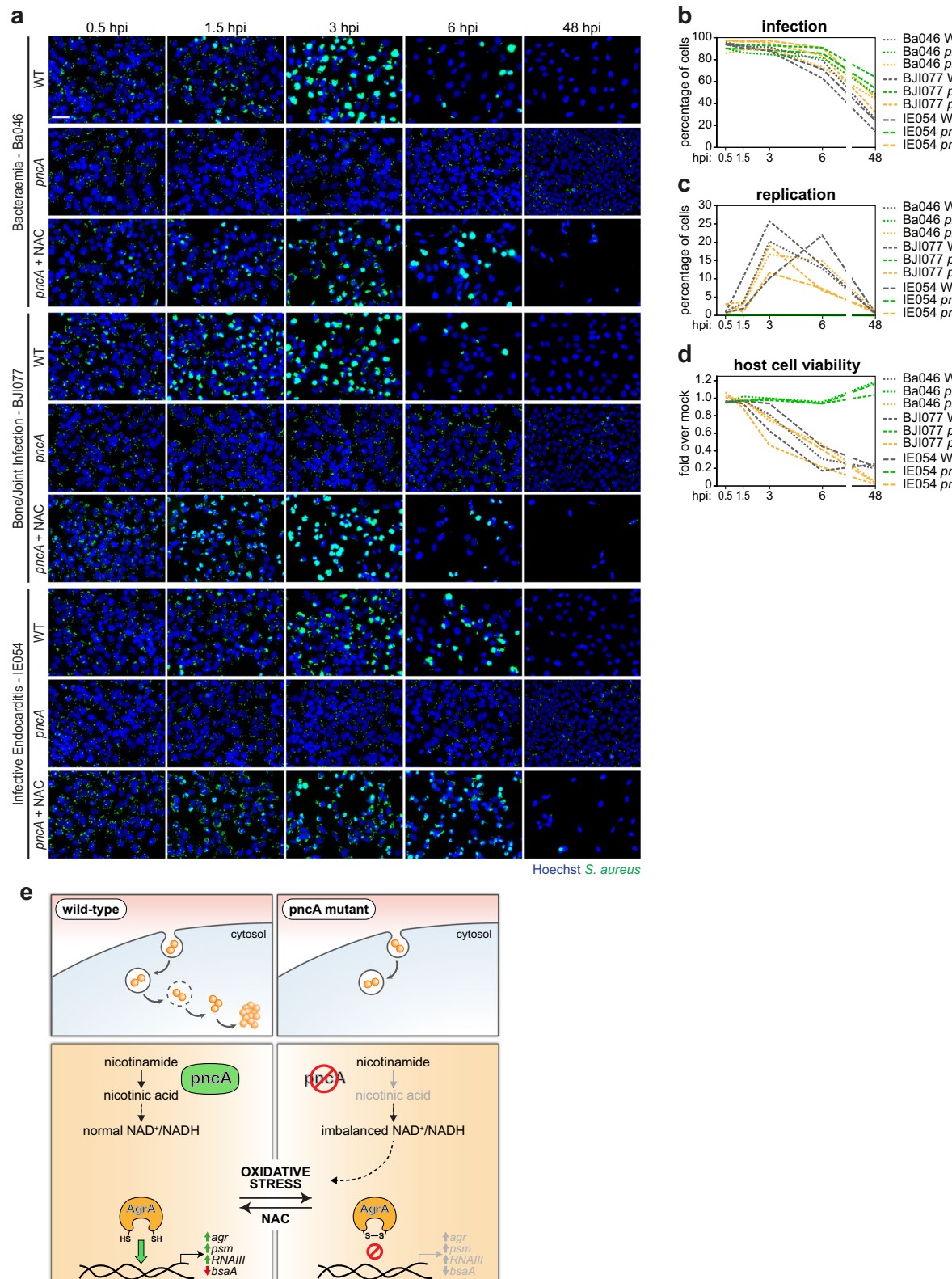

Hoechst *S. aureus*

Moreover, this study did not address the bacterial factors regulating the intracellular phenotypes in professional phagocytes, where the intracellular environment and bacterial behavior differ markedly from non-professional phagocytes. A largely different, though equally interesting, set of factors is likely to be obtained from such analysis.

The subcellular localization of *S. aureus* within infected cells significantly influences its intracellular fate. Previous studies have shown

that some *S. aureus* isolates escape the vacuole to the cytosol in non-professional phagocytes, while others remain vacuole-enclosed[18,40,50,51,63]. In general, *S. aureus* cytosolic localization correlates with high host cell death. In our analysis of the subcellular localization of the 73 selected *S. aureus* mutant strains, most strains escaped the vacuole in a manner comparable to the WT strain. Interestingly, these included mutant strains presenting low replication

**Fig. 7 | PncA regulates the intracellular profiles of *S. aureus* clinical isolates.** **a**–**d** Representative fluorescence microscopy images (**a**) and time-course analysis of infection (**b**), intracellular replication (**c**), and host cell viability (**d**) upon infection of HeLa cells with *S. aureus* clinical isolates (Ba046, BJI077, and IE054), including for each isolate the WT, the corresponding *pncA* mutant, and the *pncA* mutant supplemented with NAC (*pncA* + NAC). Results shown in panels b-d are the mean of three biologically independent experiments; microscopy images are representative of three biologically independent experiments. Scale bar, 50 μm. **e** Model depicting intracellular phenotypes of *S. aureus* wild-type (left) and *pncA* mutant (right). The pncA mutant is unable to convert nicotinamide into nicotinic acid, a precursor of $NAD^+$, leading to oxidative stress via $NAD^+$/NADH imbalance. Oxidative stress blocks AgrA association to cognate DNA, via the formation of an intramolecular disulfide bond between two cysteines in the AgrA DNA-binding domain[62]. Treatment of the *pncA* mutant with the antioxidant NAC restores the intracellular bacterial profile to that of WT strain. Source data are provided as a Source data file.

(clusters b, c, d, f, and g), suggesting the existence of bacterial factors specifically required for cytosolic replication, beyond those facilitating vacuolar escape. Interestingly, some mutants showing reduced cytosolic replication (clusters d, f, and g) were still able to induce host cell death to levels similar to those of the WT strain or even with faster kinetics (cluster c), indicating that cytosolic replication is not a prerequisite for host cell death. Importantly, we also identified strains with strongly reduced vacuolar escape, eight of each belonging to cluster h. From these, seven strains had already been associated with an impaired *agr* system and α-type PSM secretion, thus explaining the reduced escape from the vacuole. Notably, in this study, we characterized the remaining mutant strain belonging to cluster h, NE1738-*pncA*, as *agr*-defective (see below).

The NTML has been widely used to identify *S. aureus* genes involved in various processes, mostly related to resistance to antimicrobial agents and biofilm formation. Importantly, this study is the first to systematically employ the full NTML to analyze in detail the molecular correlates of bacterial intracellularity. However, the NTML has been used previously to study host cell death upon *S. aureus* infection of human neutrophils and endothelial cells[29,30], though with significantly lower resolution (single timepoint, simple host viability readouts). Although there are important intracellular phenotypic differences dependent on the cell type, particularly between professional and non-professional phagocytic cells, there is a core set of mutants that was identified in the above-mentioned screenings and in the present study—this includes mutants of core components of the *agr* system (*agrA*) and of the *agr* repressor *clpP*, presenting reduced host cell death. In addition, a subset of the NTML (144 mutants) was screened for phagosomal escape in epithelial cells[64]. From the ten strains strongly reducing vacuolar escape, four of them (NE95-*agrB*, NE873-*agrC*, NE1532-*agrA*, and NE1908-*pmtC*) were also tested in our study and similarly exhibited reduced vacuolar escape. Additionally, our screen recapitulated other previously described phenotypes, including: (i) mutants of the *agr* system affecting replication and persistence[36–38]; (ii) mutants of *rot* and *rpiRc* (virulence factor expression repressors) causing high host cytotoxicity[42,44–46]; (iii) mutant of *scpA* (encoding for the cysteine protease staphopain A) leading to high intracellular replication and delayed host cell death[10,22]; (iv) mutant of *srtA* (encoding sortase A, a key protein anchoring *S. aureus* adhesins to the cell wall peptidoglycan) displaying impaired invasion[65–67]. Overall, the consistency of these results with known *S. aureus* biology underscores the validity of our systematic approach in identifying novel factors relevant to its intracellular lifestyle. Furthermore, the grouping of uncharacterized and characterized mutants within defined phenotypic clusters will provide insights into the mechanisms of action of the neighboring uncharacterized factors.

As a case study, this work describes the identification and characterization of the nicotinamidase PncA as a novel regulator of *agr* system function (Fig. 7e). Mechanistically, we demonstrate that the *pncA* mutant exhibits elevated ROS levels, consistent with the recognized role of $NAD^+$/NADH in redox homeostasis[56,57]. This redox imbalance leads to the inactivation of the *agr* system. It has been shown that oxidative stress induces the formation of an intramolecular disulfide bond between the redox-active Cys-199 of the DNA-binding domain of the response regulator AgrA and the spatially proximate Cys-228[62]. This disrupts AgrA ability to bind its cognate DNA, resulting

in impaired regulation of downstream target genes. In line with the reduced virulence of the *pncA* mutant that we observed in both *Galleria mellonella* and mouse infection models, prior research has reported that knockdown of nicotinamide adenine dinucleotide kinase (NADK), which catalyzes the synthesis of NADP from $NAD^+$, reduces *S. aureus* induced mortality in zebrafish[68]. PncA and de novo $NAD^+$ synthesis have been implicated in the intracellular replication of other bacterial pathogens, specifically *Brucella abortus* and *Coxiella burnetii*[69,70], although the underlying mechanisms remain unidentified. Overall, our findings reveal a key role for $NAD^+$/NADH homeostasis and associated redox imbalance in modulating *S. aureus* virulence by regulating *agr* system function. Moreover, the results of *pncA* mutants generated in *S. aureus* clinical isolates demonstrate the generalizability of our findings regarding the role of PncA as a regulator of intracellular *S. aureus*, by regulating *agr* system activity.

The identification of regulators of the *agr* system remains particularly relevant, as *agr*-defective isolates have been associated with chronic and relapsing infections and identified in a variety of clinical contexts, including bacteraemia, infective endocarditis, and osteomyelitis[58,71]. While the *agr* regulon has been extensively investigated, its regulatory network is not yet fully elucidated. Notably, mutants of other reported positive regulators of the *agr* system, specifically NE1262-*mroQ*[72,73] and NE1193-*sarA*[74,75], were also identified as strong hits in our study, both exhibiting reduced intracellular replication. However, these mutants did not cluster with the other *agr*-deficient mutants (cluster h). Instead, *mroQ* and *sarA* mutants were categorized into clusters b and d, respectively. These differences in clustering may reflect variations in the extent of *agr* inhibition and/or the presence of *agr*-independent functions associated with these regulators. It is also noteworthy that genes involved in purine biosynthesis exhibited a common phenotype, i.e., reduced intracellular replication (<3-fold), yet were distributed across distinct clusters (*purA* and *purH*, cluster a; *purF*, *purL*, *purK*, cluster b; *purB*, cluster d). This likely results from the increased complexity and granularity of the phenotypic analysis. These results further reinforce the ability of our approach to capture the distinct impact that factors with seemingly similar functions can have on the intracellular behavior of *S. aureus*.

Notwithstanding the relevance of our study in identifying *S. aureus* factors relevant for intracellularity, one major limitation is the exclusion of essential genes from our analysis, arising from the inability of essential gene mutants to survive during the construction of transposon libraries. Future studies employing inducible CRISPR interference (CRISPRi) libraries, enabling the knockdown of both essential and non-essential genes[76,77], might overcome this challenge. Another limitation lies in the potential redundancy of gene functions, which may have hindered the identification of some factors critical to the observed phenotypes. For example, few metabolic genes were found among the strongest hits of the screening, likely reflecting functional redundancy. Another interesting prototypic example are the wall-associated fibronectin-binding proteins, FnBPA and FnBPB, which are key determinants of *S. aureus* adhesion and invasion of non-professional phagocytic cells, since either protein alone is sufficient to enable bacterial adhesion/invasion[15,78,79]. Consistent with this redundancy, the single mutants present in the NTML library (NE186-*fnbA* and NE728-*fnbB*) did not significantly impair *S. aureus* invasion.

In conclusion, this study provides a comprehensive identification of *S. aureus* factors critical for its intracellular lifestyle, offering a valuable resource for advancing our understanding of host–pathogen interactions. Moreover, by uncovering key players that enable *S. aureus* to invade, replicate, and persist within host cells, our findings shed light on bacterial strategies for immune evasion and intracellular survival, which might be relevant to inform the development of targeted antimicrobial interventions by identifying potential vulnerabilities in the bacterial intracellular life cycle.

## Methods

### Mammalian cell culture

Human epithelial HeLa-229 (ATCC, CCL-2.1) and human osteosarcoma U2OS (ATCC, HTB-96), were cultured in DMEM containing 1.0 g/l glucose (HyClone, SH30021.01); human endothelial EA.hy926 (ATCC, CRL-2922) cells were cultured in DMEM containing 4.5 g/l glucose (HyClone, SH30243.01); human monocyte THP1 (ATCC, TIB-202) cells were cultured in RPMI 1640 GlutaMAX (HyClone, SH30027.01). Media were supplemented with 10% fetal bovine serum (Gibco, 10270-106). Cell lines were acquired from ATCC/LGC Standards, and no further authentication was performed. Human umbilical vein endothelial cells (HUVEC; Lonza, C2519A) were maintained in EGM-2 Endothelial Cell Growth Medium-2 BulletKit (Lonza, CC-3162), according to the vendor's instructions. HeLa, U2OS, and EA.hy926 cells stably expressing mRFP-CWT were generated using the pLVTHM-H2B-BFP-IRES-mRFP-CWT plasmid (gift from M. Fraunholz). The generation of HeLa cells stably expressing mRFP-CWT has been previously described[10]; U2OS and EA.hy926 cells stably expressing mRFP-CWT were generated following the same strategy.

All cells were maintained at 37 °C in a humidified atmosphere with 5% $CO_2$. All cell lines tested negative for mycoplasma contamination.

### Bacterial strains and growth profiles

*S. aureus* USA300 JE2 is a derivative of USA300 LAC, a community-associated MRSA, which was cured of three plasmids. The USA300 JE2 (NR-46543), used to generate the Nebraska Transposon Mutant Library (NTML), the NTML (NR-48501), and *S. aureus* strain RN6911 (NR-45953), were provided by the Network on Antimicrobial Resistance in *Staphylococcus aureus* (NARSA) through BEI Resources, NIAID, NIH (www.beiresources.org). The NTML arrayed in five 384-well microtiter plates was reformatted into 22 round-bottom 96-well plates (Corning, 3799), by transferring the bacterial stocks using 96-pin replicators (Scinomix, SCI-4010-OS) to 100 µl of Trypticase Soy broth (TSB; BD Biosciences, 211771) supplemented with erythromycin 5 µg/ml. Bacteria were grown overnight at 37 °C with shaking. DMSO was then added to a final concentration of 10%, and the plates were subsequently stored at −80 °C. These plates were used as stock plates for the infection assays. Three strains (NE235, mutant for SAUSA300_RS04575; NE352-*rsgA*, mutant for SAUSA300_RS06030; NE1896-*lpdA*, mutant for SAUSA300_RS08010) did not grow in TSB after several attempts and were not considered for further analysis.

The location of the transposon insertion for the NE1738-*pncA* mutant strain was validate by PCR using the following primer pair 5′-CTCGATTCTATTAACAAGGG-3′ and 5′-GCAAGTGCCCATTCATGCCC-3′, which anneal to *bursa aurealis* transposon and *pncA* gene, respectively. Briefly, overnight bacterial cultures were pelleted, and the chromosomal DNA was extracted by lysing the cells in lysis buffer (20 mM Tris pH 7.5, 10 mM EDTA, 25 µg/ml lysostaphin) and incubating for 10 min at 37 °C, followed by DNA extraction by phenol:chloroform:isoamyl alcohol (25:24:1; Invitrogen, 15593-031) followed by isopropanol precipitation. Purified DNA samples were used as template with the aforementioned primers.

To generate the *pncA* complemented strain, the *pncA* gene, including the native promoter region (144 bp upstream of the start codon), was amplified by PCR from *S. aureus* genomic DNA (*S. aureus*

USA300 JE2) using the following primer pair 5′-AAACTGCAG-GATTGCGTGATTTGCTCTTCC-3′ and 5′-GGCGGATCCT-TAAACGTGTTGTTCTACCTCTGC-3′. The product was cloned in the pjL74 plasmid using the PstI and BamHI restriction sites. The resulting plasmid was passaged through *E. coli* IM08B plasmid artificial modification strain[80] (gift from I. Monk), and the purified plasmid was transformed into the NE1738-*pncA* mutant strain by electroporation. Transformants were checked by colony PCR and verified by DNA sequencing.

The NE1738-*pncA* transposon insertion was transferred by Phi11 phage transduction[81] into three *S. aureus* clinical isolates obtained from patients with bacteraemia, bone/joint infections, and infective endocarditis[82,83], generating the corresponding *pncA* mutant strains. The intracellular profiles of the parental isolates had been characterized previously[10]. In addition, the *S. aureus* strain RN6911, in which the *agr* operon was replaced with tetracycline resistance gene[84], was used as the donor strain for Phi11 phage transduction into the NE1738-*pncA* strain, yielding a *pncA/agr* mutant strain.

For generating the growth curves, overnight bacterial cultures were diluted 1:100 and grown in 100 µl TSB, RPMI, or RPMI supplemented with nicotinic acid (NA; 0.3 mM; Sigma, 72309-1006) for 8 h at 37 °C in round-bottom 96-well plates, under continuous orbital shaking. $OD_{600}$ was measured using an EnSpire Multimode plate reader (PerkinElmer), at 30 min intervals.

### Fluorescence microscopy-based infection assays

For the time-course assays, cells were seeded in black, clear-bottom 384-well plates (Greiner, 781090). HeLa, U2OS, EA.hy926, and HUVEC cells were plated 48 h before infection, at a density of $1.5 \times 10^3$ (HeLa), $1.6 \times 10^3$ (U2OS, EA.hy926), or $1.25 \times 10^3$ (HUVEC) cells per well. *S. aureus* mutant strains (NTML or 73 selected isolates) were grown overnight in round-bottom 96-well plates (Corning, 3799) at 37 °C with shaking. Overnight bacterial growths were diluted 1:100 in TSB and grown for 2 h at 37 °C with shaking, allowing the majority of bacterial cultures to reach $OD_{600}$ 0.38–0.85, median 0.51 (exponential phase). For slow-growing mutants, strains were grown for additional 15–30 min intervals to ensure that all bacterial isolates were within the indicated range. $OD_{600}$ was determined using an EnSpire Multi-mode Plate Reader (PerkinElmer). Infections were performed at a multiplicity of infection (MOI) ~25 by adding 10 µl of bacterial suspensions previously diluted 1:32 in complete medium for the mammalian cells. Cells were centrifuged at $500 \times g$, at room temperature (RT) for 10 min, and incubated at 37 °C in a 5% $CO_2$ humidified atmosphere for 50 min. Extracellular bacteria were killed by replacing the medium with fresh medium containing 5 µg/ml of the bacteriolysin lysostaphin (Ambi Products LLC, LSPN-50), which was defined as timepoint 0 h. Medium supplemented with the bacteriolysin was maintained until the indicated times of collection. USA300 JE2 (WT), NE1278-*scpA*, and NE1532-*agrA* were used for standardization of the assay and included in all assay plates.

For *S. aureus* vacuolar escape, HeLa, U2OS, and EA.hy926 cells stably expressing mRFP-CWT were seeded 48 h before infection in black, clear-bottom 384-well plates at a density of $1.5 \times 10^3$ (HeLa) or $1.6 \times 10^3$ (U2OS, EA.hy926) cells per well. Dilution of bacterial cultures and infections was performed as described above.

For the experiments with *S. aureus pncA* mutant, complemented strain, and nicotinic acid (NA) or N-acetyl-L-cysteine (NAC; Sigma, A7250) treatment experiments, cells were plated as described above. When indicated, cells were treated with 0.3 mM NA or 25 mM NAC 30 min prior to infection and during infection. *S. aureus* overnight cultures were diluted 1:100 in TSB and grown at 37 °C with shaking until $OD_{600}$ ~0.4; when indicated, bacterial growth was performed in TSB supplemented with NA (0.3 mM) or NAC (25 mM). Bacteria were harvested by centrifugation for 2 min at $12,000 \times g$ and resuspended in complete medium. Infections were performed as described above.

THP1 cells were plated 72 h before infection, at a density of $6 \times 10^3$ cells per well. THP1 cells were differentiated into macrophage-like cells by treatment with 50 ng/ml phorbol 12-myristate 13-acetate (PMA, Sigma, 79346) at the time of plating and incubated with PMA for 72 h, without a resting period. Infections were performed as described above.

For the eFluor proliferation assay, labelling of *S. aureus* with eFluor-670 cell proliferation dye was performed as previously described[10]. Briefly, bacteria were grown as described above, followed by labeling with 0.5 μg/ml eFluor-670 dye (eBiosciences, 65-0840-90) in PBS for 5 min at RT. Bacteria were then harvested by centrifugation at $12,000 \times g$ for 2 min, and resuspended in LB for 3 min to quench unreacted dye. Bacteria were washed twice with PBS and then resuspended in complete medium. Infections with the labeled bacteria were performed as described above.

### Staining and immunofluorescence
At the indicated times post-infection, cells seeded in black, clear-bottom 384-well plates were washed with PBS and fixed with 4% paraformaldehyde for 15 min at RT or overnight at 4 °C (HeLa, U2OS, and EA.hy926 cells stably expressing mRFP-CWT). After 30 min of permeabilization with 0.5% Triton X-100 (Roth, 3051.4) in PBS, *S. aureus* was labeled for 2 h at RT with 0.25 μg/ml BODIPY-FL vancomycin (Invitrogen, V34850), a glycopeptide antibiotic conjugated to a fluorescent dye that specifically binds to the cell wall of Gram-positive bacteria[34,35]. Cells were washed, and nuclei were counterstained with Hoechst 33342 (1:5,000; Thermo Fisher Scientific, H3570) for 15 min at RT. For staining of acidic vesicles, cells were treated for 30 min at 37 °C prior to fixation with 75 nM LysoTracker Red DND-99 (Life Technologies, L-7528). Plates were kept at 4 °C until image acquisition. To ensure reproducibility and speed, cell washing in 384-well plates was performed using an automated Bio-Tek ELx405 Multiplate Washer (BioTek), and reagent addition was performed using an automated Multidrop Combi Reagent Dispenser (Thermo Fisher Scientific).

### Image acquisition and analysis
Image acquisition was performed using an Operetta automated high-content screening fluorescence microscope (PerkinElmer), at 20× magnification, with a total of nine images acquired per well. Image analysis to quantify *S. aureus* infection, intracellular replication, host cell viability, bacterial vacuolar escape (*S. aureus*/CWT colocalization, and *S. aureus*/LysoTracker colocalization) was performed using custom workflows implemented in Harmony/Columbus image analysis software (PerkinElmer), as previously described in ref. 10,85. Briefly, to quantify *S. aureus* infection and replication, the first step was segmentation of the nucleus, cytoplasm, and bacteria based on a combination of global and individual thresholds and other parameters (e.g., split factor, area, and contrast). This was followed by extraction of intensity and morphological features (including integrated intensity and area) and classification of cells as infected/non-infected or with high bacterial replication (cells with high intracellular *S. aureus* load) by applying cutoff criteria based on the extracted features. For the evaluation of *S. aureus* vacuolar escape using the CWT reporter and LysoTracker, after the selection of the infected cells as described above, mRFP-CWT or LysoTracker spots were identified within the cytoplasm of infected cells, and the area of mRFP/CWT or LysoTracker spots were then restricted to the area of *S. aureus*, and finally these areas were used to calculate *S. aureus*/mRFP-CWT or *S. aureus*/LysoTracker colocalization ratios. For all the analyses, cells touching the edge of the images were excluded.

### Colony-forming unit assays
HeLa cells were seeded in 24-well plates (Corning, 3526) at a density of $5 \times 10^4$ cells per well 48 h before infection. *S. aureus* overnight cultures were diluted 1:100 in TSB and grown at 37 °C with shaking until $OD_{600}$ 0.4. Bacteria were then harvested by centrifugation for 2 min at $12,000 \times g$ and resuspended in complete medium for mammalian cells. Infections with the indicated *S. aureus* strains were performed at MOI 25, as described above in "Fluorescence microscopy-based infection assays". At the indicated times post-infection, cells were washed with PBS and lysed with PBS containing 0.1% Triton X-100. Lysates were serially diluted in PBS and plated on TSB agar plates.

### Mammalian cell viability assay
HeLa cells were seeded in black, clear-bottom 384-well plates (Greiner, 781090) 48 h before infection at a density of $1.5 \times 10^3$ cells per well. Bacterial growth and infections were performed as described above in "Fluorescence microscopy-based infection assays". Host cell viability was determined using the ATPLite 1-step Luminescence Assay System (PerkinElmer, 6016731), according to the manufacturer's instructions, and luminescence was measured using an EnSpire Multimode Plate Reader (PerkinElmer).

### Measurement of haemolytic activity
For plate-based haemolysis assays, 3 μl of *S. aureus* cultures (overnight cultures diluted at 1:100 and grown at 37 °C with shaking for 8 h) were spotted onto tryptic soy agar (TSA) containing 2% defibrinated sheep blood (Thermo Fisher Scientific, PB5012A). The plates were incubated at 37 °C and were imaged at 6 and 24 h.

Quantitative haemolysis assays of *S. aureus* culture supernatants were performed essentially as described previously[86]. Briefly, overnight bacterial cultures were diluted at 1:100 in 10 ml of TSB and grown at 37 °C with shaking for 8 h. Bacterial cultures were centrifuged ($12,000 \times g$ for 5 min) and the supernatant was collected and filtered through a 0.2-μm filter. Serial 2-fold dilutions of the supernatants in TSB (100 μl) were mixed with 2% sheep blood suspension in PBS (100 μl; Thermo Fisher Scientific, SR0053B) in clear, round-bottom 96-well plates (Corning, 3799), and incubated statically at 37 °C for 30 min. Unlysed blood cells were removed by centrifugation ($1,700 \times g$ for 5 min), and the supernatants were transferred to new 96-well plates, followed by measurement of absorbance at 405 nm using an EnSpire Multimode plate reader (PerkinElmer). Blood incubated with TSB or TSB containing 1% Triton X-100 served as negative and positive controls, respectively. All values are presented normalized to $OD_{405}$ of the positive control (TSB + 1% Triton).

### NAD+ & NADH quantification
Total levels of NAD+ & NADH in *S. aureus* strains were determined using the NAD+/NADH-Glo assay kit (Promega, G9071), according to the manufacturer's instructions. Briefly, *S. aureus* overnight cultures were diluted 1:100 in TSB and grown at 37 °C with shaking for 8 h. 25 μl of the bacterial culture was mixed and incubated with 25 μl of NAD/NADH-Glo detection reagent in white 384-well plates (Corning, 3570), followed by gentle mixing and incubation for 30 min at RT. Luminescence was measured using an EnSpire Multimode Plate Reader (PerkinElmer).

### Oxidative stress (ROS levels) quantification
Bacterial ROS levels were determined using the CM-H2DCFDA probe. Briefly, overnight bacterial cultures were diluted 1:100 in 10 ml of TSB and grown at 37 °C with shaking until $OD_{600}$ ~0.4. Bacterial cultures (100 μl) were harvested by centrifugation for 5 min at $10,000 \times g$ and washed three times with PBS. The bacterial suspensions were then resuspended in 100 μl of PBS containing 5 μM CM-H2DCFDA (Thermo Fisher, C6827) and incubated for 30 min at RT protected from light. The labeled cells were washed once with PBS, resuspended in 100 μl of PBS, and transferred to black, clear-bottom 96-well plates (Greiner, 655090). The fluorescence signal (excitation 495 nm, emission 520 nm) was measured using an EnSpire Multimode Plate Reader (PerkinElmer).

## RNA isolation, cDNA synthesis, and quantitative real-time PCR

RNA extraction, cDNA synthesis, and qPCR were performed as described previously[10]. Briefly, for *S. aureus* cultures grown to stationary phase (overnight cultures diluted at 1:100 and grown at 37 °C with shaking for 8 h), samples were centrifuged and lysed in TRIzol (Invitrogen, 15596026). For infected epithelial cell samples, HeLa cells were seeded in 24-well plates at a density of $5 \times 10^4$ cells per well 48 h before infection and infected as described above "Fluorescence microscopy-based infection assays" (overnight cultures diluted 1:100 in TSB and grown at 37 °C with shaking until $OD_{600}$ ~0.4; MOI 25). At 3 hpi, cells were lysed in TRIzol.

RNA was extracted by phenol-chloroform followed by isopropanol precipitation, and used for cDNA synthesis using hexameric random primers and M-MLV reverse transcriptase (Invitrogen, 28025021) according to the manufacturer's instructions. qRT-PCR was performed using Sso Advanced Universal SYBR Green Supermix (Bio-Rad, 172-5274) according to the manufacturer's instructions, using a CFX96 TouchTM Real-Time PCR detection system (Bio-Rad). The following primer pairs were used: RNAIII 5′-GAAGGAGTGTTTCAATGG-3′ and 5′-TAAGAAAATACATAGCACTGAG-3′[87]; PSM-α 5′-GGCCATTCACATGGAA TTCGT-3′ and 5′-GCCATCGTTTTGTCCTCCTG-3′[18]; AgrA 5′-AT GGAAATTGCCCTCGCAA-3′ and 5′-CCAACTGGGTCATGCTTACGA-3′[18]; BsaA 5′-GTGTTGTGCCGCAGTCAAAT-3′ and 5′-ACCTGGTTCAGG CGAAGAAG-3′; GAPDH 5′-TACACAAGACGCACCTCACAGA-3′ and 5′-ACCTGTTGAGTTAGGGATGATGTTT-3′[88]. Expression was normalized to GAPDH, and the relative gene expression was calculated using the $2^{-\Delta\Delta Ct}$ method.

## Metabolite extraction experiments

For the targeted metabolomics profiling studies, *S. aureus* overnight cultures were diluted 1:100 in TSB and grown at 37 °C with shaking for 8 h. Bacterial cultures (10 ml) were harvested by centrifugation for 5 min at $10,000 \times g$, washed three times with cold PBS, and suspended into the extraction solution composed of acetonitrile/methanol/$H_2O$ (2:2:1), precooled to 4 °C. Small molecules were extracted by the mechanical lysis of the entire bacteria-containing solution with 0.1-mm acid-washed zirconia beads for 1 min using a FastPrep system (MP Bio) set to 6.0 m/s. The lysates were filtered twice through 0.22-μm Spin-X column filters (Costar, 8161). The bacterial biomass of the individual samples was determined by measuring the residual protein content of the metabolite extracts using the BCA protein assay kit (Thermo Fisher, 23225)[89,90]. A 100-μl aliquot of the metabolite solution was then mixed with 100 μl of acetonitrile with 0.2% acetic acid at −20 °C, and the mixture was centrifuged for 10 minutes at $17,000 \times g$ at 4 °C. The final concentration of 70% acetonitrile was compatible with the starting conditions of Diamond Hydride chromatography. The supernatant was then transferred into LC/MS V-shaped vials (Agilent 5188–2788), and a 4 μl aliquot was injected into the LC/MS instrument.

## Liquid chromatography-mass spectrometry for targeted metabolomic analysis

Aqueous normal-phase liquid chromatography was performed using an Agilent 1290 Infinity II LC system equipped with a binary pump, temperature-controlled autosampler (set at 4 °C), and temperature-controlled column compartment (set at 25 °C) containing a Cogent Diamond Hydride Type C silica column (150 mm × 2.1 mm; dead volume of 315 μl). A flow rate of 0.4 mL/min was used. The elution of polar metabolites was performed using solvent A, which consisted of deionized water (resistivity ~18 MΩ cm) and 0.2% acetic acid, and solvent B, which consisted of 0.2% acetic acid in acetonitrile. The following gradient was applied at a flow rate of 0.4 ml/min: 0 min, 85% B; 0–2 min, 85% B; 3–5 min, 80% B; 6–7 min, 75% B; 8–9 min, 70% B; 10–11 min, 50% B; 11.1–14 min, 20% B; 14.1–25 min, 20% B; and 5−min of re-equilibration at 85% B. Accurate mass spectrometry was performed using an Agilent Accurate Mass 6545 QTOF apparatus. Dynamic mass axis calibration was achieved by continuous infusion after the chromatography of a reference mass solution using an isocratic pump connected to an ESI ionization source operated in positive-ion mode. The nozzle and fragmentor voltages were set to 2000 V and 100 V, respectively. The nebulizer pressure was set to 50 psig, and the nitrogen drying gas flow rate was set to 5 L/min. The drying gas temperature was maintained at 300 °C. The MS acquisition rate was 1.5 spectra/s, and $m/z$ data ranging from 50 to 1200 were stored. This instrument enabled accurate mass spectral measurements with an error of less than 5 parts per million (ppm), a mass resolution ranging from 10,000 to 45,000 over the $m/z$ range of 121–955 atomic mass units, and a 100,000-fold dynamic range with picomolar sensitivity. The data were collected in centroid 4-GHz (extended dynamic range) mode. The detected $m/z$ data were deemed to represent metabolites, which were identified based on unique accurate mass-retention times and MS/MS fragmentation identifiers for masses exhibiting the expected distribution of accompanying isotopomers. The typical variation in the abundance of most metabolites remained between 5 and 10% under these experimental conditions.

## Nicotinic acid standard curve

A stock solution of nicotinic acid at a concentration of 20 μg/mL in double-distilled water was prepared and serially diluted in a solution of acetonitrile/methanol/$H_2O$ (2:2:1) to obtain concentrations in the range of 20 μg/mL to 2 ng/mL in technical duplicates. A standard curve was established using Agilent Quantitative Analysis B.07.00.

## *Galleria mellonella* infection

Final instar larval stage *G. mellonella* were purchased from UK Waxworms. Groups of 10 larvae were randomly assigned (active larvae with no visible melanization and weighing 250–300 mg), and individual larvae were infected with $2 \times 10^6$ CFU of *S. aureus* by intrahemocelic injection in the last left proleg. *S. aureus* inoculum was prepared as described above and diluted in PBS. After injection, infected larvae were incubated at 37 °C, and the survival of each group was scored daily for 7 days. The control group was inoculated with vehicle (PBS), with a maximum of one dead larva observed by day 7 (not included in Fig. 6b for simplicity). Larvae were considered dead when displaying melanization and showing no reaction in response to gentle tapping. Four independent experiments were performed with 10 larvae tested per strain and experiment.

## Mouse pneumonia infection model

Mice (*Mus musculus* BALB/c OlaHsd, 8 weeks old, body weight 16–19 g) were purchased from Envigo and were housed in polypropylene cages under standardized lighting conditions with ad libitum access to food and water. For CFUs, animals ($n = 11$ per group; 8–10 weeks old) were infected with $2 \times 10^8$ CFU of *S. aureus* via nasal instillation. Infection was allowed to progress for 24 h, the mice were then sacrificed, and lungs collected aseptically, homogenized, and plated in TSB plates for CFU counting. For survival, mice ($n = 8$ per group; 8 weeks old) were infected with $6 \times 10^8$ CFU, and survival was recorded over 4 days. CFU experiments were conducted in both female and male mice, with no observed differences (results shown are the combined data); survival experiments were performed in female mice only, as our previous studies with high infection doses showed no sex-dependent differences in survival.

All animal studies were approved by the Madrid Regional Government, Spain (license no. PROEX 049.3-24) and performed in strict accordance with the guidelines for animal care and experimentation according to Spanish law and EU directive 2010/63/EU.

## Statistical/data analysis

Unless otherwise indicated, data are presented as mean or mean ± standard error of the mean (s.e.m.), with the exact number of

independent experiments performed indicated in figure legends. Microsoft Excel (Microsoft 365) was used to compile the data, and Prism software (v9.3.0; GraphPad) was used to generate the graphs and perform statistical analysis. The normal distribution of the data was assessed by the Shapiro–Wilk test. For the box-plots (Tukey method), black lines show the medians, box limits indicate the 25th-75th percentiles, whiskers extend 1.5 times the interquartile range from the 25th and 75th percentiles; when only three values per group were included, the plot shows the median and range. The following tests were used for statistical comparison of datasets: from three or more groups, one-way ANOVA with Dunnett's post hoc test (for parametric data), or Kruskal–Wallis with Dunn's multiple comparison test (non-parametric data), or two-way ANOVA with Dunnett's post hoc test; for the survival curves, log-rank (Mantel–Cox) test. Values of $P < 0.05$ were considered significant. Statistical analyses are detailed in Supplementary Data 4. Violin plots were generated using BoxPlotR web tool (http://shiny.chemgrid.org/boxplotr/); white circles show the medians, box limits indicate the 25th–75th percentiles, whiskers extend 1.5 times the interquartile range from the 25th and 75th percentiles, and polygons extend to extreme values. Heat map of the phenotypic profiles of the 73 selected *S. aureus* mutants and hierarchical clustering (method UPGMA, Euclidean distance) was performed using Spotfire (v.14.0.2; Cloud Software Group, Inc). Before clustering, each phenotype (infection, replication, host cell viability) across the 3 cell lines was normalized to the highest value therein. Adobe Photoshop (v26.3.0; Adobe) and Adobe Illustrator (v29.2.1.; Adobe) were used for assembling microscopy images and figures, respectively.

### Reporting summary

Further information on research design is available in the Nature Portfolio Reporting Summary linked to this article.

## Data availability

The data corresponding to the *S. aureus* phenotypic screening, follow-up experiments with 73 selected *S. aureus* strains in various cell lines and vacuolar escape are provided in Supplementary Data 1-3. Raw microscopy datasets have been deposited to BioImage Archive, accession number S-BIAD2374. Custom image analysis workflows implemented in Harmony/Columbus image analysis software (PerkinElmer) are available from the corresponding authors upon request. Metabolomics data have been deposited to MetaboLights repository with the study identifier MTBLS13220, available at https://www.ebi.ac.uk/metabolights/MTBLS13220. Source data are provided with this paper.

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

## Acknowledgements

I.R.L. was a recipient of a PhD fellowship (PD/BD/146464/2019) of the Doctoral Programme in Experimental Biology and Biomedicine of the Center for Neuroscience and Cell Biology, University of Coimbra. We thank Martin J. Fraunholz (University of Würzburg, Germany) for providing the pLVTHM-H2B-BFP-IRES-mRFP-CWT plasmid, Ian Monk (University of Melbourne, Australia) for the *E. coli* IM08B strain, and Frederic Laurent and Francois Vandenesch (Centre National de Référence des Staphylocoques, Institut des Agents Infectieux, Hospices Civils de Lyon, France) for the *S. aureus* clinical isolates. This work was supported by grants from: ERA-NET Infect-ERA StaphIN (031L0094, BMBF, Germany, to A.E.; Infect-ERA/0001/2015, FCT, Portugal, to M.M.; PCIN-2015-151, MINECO, Spain, to D.L); European Regional Development Fund (ERDF), through the Centro 2020 Regional Operational Program, and through the COMPETE 2020—Operational Programme for Competitiveness and Internationalization, and Portuguese national funds via FCT under the projects UIDB/04539/2020, UIDP/04539/2020, LA/P/0058/2020 to M.M. and A.E.; Associação AccelBio, Collaborative Laboratory to Foster Translation and Drug Discovery, and funds from Agência Nacional de Inovação and European Union (REC05-i02 –N° 01/C05-i02/2022) to M.M. Open access fee was paid from the Imperial College London Open Access Fund.

## Author contributions

I.R.L. performed and analyzed the data from the *S. aureus* NTML screening, the *S. aureus* mutants phenotypic and mechanistic characterization, and participated in manuscript writing; L.M.A. performed *S. aureus* mutants phenotypic and mechanistic characterization; M.L.B. and D.L. performed the in vivo experiments with the mouse pneumonia model of *S. aureus* infection; Y.L. and G.L.M. performed the targeted metabolomic analysis for nicotinic acid quantification; M.M. and A.E. coordinated the work and wrote the manuscript, with input from all the authors.

## Competing interests

The authors declare no competing interests.
