## [Transparent Peer Review file · Nature Communications]

Systematic identification of bacterial factors driving *Staphylococcus aureus* intracellular lifestyle in non-professional phagocytes

Corresponding Author: Dr Ana Eulalio

Version 0:

Reviewer comments:

Reviewer #2

(Remarks to the Author)

The experiments delineated in this revised manuscript very convincingly document the role of the nicotinamidase PncA in the survival of *S. aureus* within a variety of relevant cell types and in clinical isolates. Its link to agr expression is logical and helps to support its importance. Less convincing are the statements that the 73 genes identified using the HeLa screen of the Tn library are a “comprehensive” list of the factors important in the intracellular life style of *S. aureus*.

Specific Issues -

1 - As pointed out by the authors, the 48 hour time point is clearly limiting – and fails to account for bacterial adaptation to the metabolic demands of the intracellular environment, which is key to their success as intracellular pathogens. While the importance of PncA in clinical isolates is well appreciated – validation of this individual locus while very convincingly performed – does not exclude the participation of many other genes in adaptation.

2-No metabolic or regulatory genes are identified using this methodology. This seems highly unlikely and further points out the limitations of this technique and the 48 hour time point.

3- This study focused upon non-immune cells. It would be important to discuss how the intracellular persistence of *S. aureus* in immune cells – and avoidance of the killing by the phagolysosome may select for different “essential” genes.

Reviewer #4

(Remarks to the Author)

In this revised manuscript by Lopes et al., the authors make a serious and commendable effort to address the original reviewers’ comments. I applaud the rigorous additional experiments, especially the validation of PncA phenotypes in THP-1 macrophages. Using clinical isolates with distinct agr phenotypes strengthens the paper and reinforces the importance of PncA. Direct measurements of nicotinic acid (NA) in the pncA mutant add mechanistic rigor to this story.

I do not think additional experiments are required prior to publication in Nature Communications. Below are suggested text edits and citations to improve clarity and context.

1) Lines 53–54: The statement that transposon mutant libraries are powerful tools is correct, but there are no citations here. To help a broad audience (like the one for Nature Comms), please add a few references for context, prioritizing large-scale bacterial Tn-seq studies involving mammalian cells and/or in vivo studies, including:

- DOI: 10.1016/j.cell.2025.02.010: Large-scale Tn-seq in *Bifidobacterium breve*, including in mice.
- DOI: 10.1038/s41588-024-01779-7: Large-scale Tn-seq in *Salmonella*, including within human macrophages.
- DOI: 10.1038/s41586-018-0124-0: Seminal multi-species Tn-seq resource paper.
- DOI: 10.1038/s41588-025-02218-x: *Shigella* transposon screens in human intestinal organoids.
- DOI: 10.1038/s41564-025-01975-z: Inducible Tn-seq with enteric pathogens in mice
- *S. aureus* with mammalian cells: DOI: 10.1128/iai.00228-23; 10.1038/s41467-025-62292-x.

2) Lines 57–59: Please state explicitly that the Nebraska Transposon Mutant Library (NTML) comprises 1,920 individually mapped mutants stored in ARRAYED format in the introduction. Clarifying this will highlight a key strength of this paper's approach: arrayed screening enables higher-resolution phenotyping in mammalian cell infections than pooled liquid libraries typically allow (e.g., image-based single-cell readouts across multiple time points). Non *S. aureus* readers may not realize that NTML is arrayed at first, as most Tn-seq libraries are pooled in liquid.

3) The paper's claim of novelty is justified for *S. aureus*; however, similar high-throughput, single-cell fluorescence microscopy approaches have been used to study bacteria-host cell interactions in other species. To more precisely define the novelty of this work, perhaps the authors could add a sentence like the following: "Where prior studies applied arrayed fluorescent promoter-based libraries in Gram-negative bacteria, our work applies high-resolution single-cell microscopy to an arrayed transposon library in a clinically important Gram-positive pathogen."

Also please cite papers in other species that have used similar techniques, including:

- DOI: 10.1038/s41564-025-01953-5 :~3,000 arrayed fluorescent reporter strains of Salmonella to profile host-pathogen interactions in macrophages.
- DOI: 10.1038/nmeth895 :Comprehensive fluorescent reporter library in Escherichia coli for high-throughput based single cell imaging.

4) As reviewers 2 and 3 have already pointed out, please continue to soften the language around "persistence", as 48 hours post infection in vitro is not true persistence. Perhaps adding a statement in the Discussion noting that definitive persistence will require longer-term or in vivo models optimized for chronic infection would be useful here.

5) In lines 508-510, please add a little bit more technical detail into how THP1 differentiation was done, as this can vary widely across labs. For example, how long were cells differentiated for? Was there a resting period?

6) I really like that the authors added some data from clinical isolates in Fig. 7. Perhaps to visually highlight the diversity of these isolates, the authors could label in Fig 7 (or add a small schematic in the Fig) that these isolates come from patients with bacteremia, bone/joint infections, and infective endocarditis, respectively. This would visually make this conserved phenotype very striking.

Systematic identification of bacterial factors driving *Staphylococcus aureus* intracellular lifestyle in non-professional phagocytes

Manuscript Number: NCOMMS-25-68706-T

Reviewers Comments: black

Our reply: blue

We are very grateful to the Reviewers for their supportive evaluation of our work and constructive comments.

Reviewer #2 (Remarks to the Author):

The experiments delineated in this revised manuscript very convincingly document the role of the nicotinamidase PncA in the survival of *S. aureus* within a variety of relevant cell types and in clinical isolates. Its link to agr expression is logical and helps to support its importance. Less convincing are the statements that the 73 genes identified using the HeLa screen of the Tn library are a “comprehensive” list of the factors important in the intracellular life style of *S. aureus*.

We would like to clarify that we do not claim that the 73 genes identified in the screening represent a comprehensive list of factors involved in the intracellular lifestyle of *S. aureus*. Throughout the manuscript, the term comprehensive is used to refer to our experimental approach, reflecting the broad scope of the screening performed, or to the full set of the genes identified, rather than the subset of 73 genes selected for follow-up. These 73 genes constitute the subset of genes chosen for further validation and characterization from the 307 mutants that exhibited alterations in at least one of the phenotypic characteristics evaluated (ca. 16%, considering a 2-fold threshold relative to control; lines 322-323).

Specific Issues -

1 - As pointed out by the authors, the 48 hour time point is clearly limiting – and fails to account for bacterial adaptation to the metabolic demands of the intracellular environment, which is key to their success as intracellular pathogens. While the importance of PcnA in clinical isolates is well appreciated – validation of this individual locus while very convincingly performed – does not exclude the participation of many other genes in adaptation.

We fully agree with the Reviewer that the 48 hpi time point is limiting to study persistence and does not capture the full spectrum of bacterial adaptations to the intracellular environment. We have reinforced this point in the revised manuscript (lines 100-102, 332-333), acknowledging that longer-term analyses would be required to comprehensively assess the contribution of additional factors beyond PncA (and the other genes in cluster h) to intracellular persistence.

2-No metabolic or regulatory genes are identified using this methodology. This seems highly unlikely and further points out the limitations of this technique and the 48 hour time point.

Our screen did identify both regulatory genes (including *agr* system, *sarA*, *rot*, *rpiRc*) and metabolic genes as relevant for the *S. aureus* intracellular lifestyle. However, as explained in the previous revision, many of the metabolic genes did not pass the stringent criteria applied for inclusion among the 73 genes selected for follow-up analyses. One possible explanation is functional redundancy, which may mask the phenotypic effects of some mutations under the tested conditions. This clarification has been included in the revised manuscript (lines 413-414).

3- This study focused upon non-immune cells. It would be important to discuss how the intracellular persistence of *S. aureus* in immune cells – and avoidance of the killing by the phagolysosome may select for different “essential” genes.

We fully agree with the Reviewer’s point and also anticipate that the sets of genes important for the intracellular lifestyle of *S. aureus* are very likely to differ between immune cells and non-professional phagocytes. This distinction has been explicitly addressed in the revised manuscript (lines 334-337).

Reviewer #4 (Remarks to the Author):

In this revised manuscript by Lopes et al., the authors make a serious and commendable effort to address the original reviewers’ comments. I applaud the rigorous additional experiments, especially the validation of PncA phenotypes in THP-1 macrophages. Using clinical isolates with distinct *agr* phenotypes strengthens the paper and reinforces the importance of PncA. Direct measurements of nicotinic acid (NA) in the *pncA* mutant add mechanistic rigor to this story.

I do not think additional experiments are required prior to publication in Nature Communications. Below are suggested text edits and citations to improve clarity and context.

1) Lines 53–54: The statement that transposon mutant libraries are powerful tools is correct, but there are no citations here. To help a broad audience (like the one for Nature Comms), please add a few references for context, prioritizing large-scale bacterial Tn-seq studies involving mammalian cells and/or in vivo studies, including:

- DOI: 10.1016/j.cell.2025.02.010: Large-scale Tn-seq in *Bifidobacterium breve*, including in mice.
- DOI: 10.1038/s41588-024-01779-7: Large-scale Tn-seq in *Salmonella*, including within human macrophages.
- DOI: 10.1038/s41586-018-0124-0: Seminal multi-species Tn-seq resource paper.
- DOI: 10.1038/s41588-025-02218-x: *Shigella* transposon screens in human intestinal organoids.
- DOI: 10.1038/s41564-025-01975-z: Inducible Tn-seq with enteric pathogens in mice
- *S. aureus* with mammalian cells: DOI: 10.1128/iai.00228-23; 10.1038/s41467-025-62292-x.

We thank the Reviewer for bringing this issue to our attention. We have added references to three recent reviews to emphasize the relevance of transposon mutant libraries in elucidating the genetic determinants of pathogenicity, including of *S. aureus*.

2) Lines 57–59: Please state explicitly that the Nebraska Transposon Mutant Library (NTML) comprises 1,920 individually mapped mutants stored in ARRAYED format in the introduction. Clarifying this will highlight a key strength of this paper’s approach: arrayed screening enables higher-resolution phenotyping in mammalian cell infections than pooled liquid libraries typically allow (e.g., image-based single-cell readouts across multiple time points). Non *S. aureus* readers may not realize that NTML is arrayed at first, as most Tn-seq libraries are pooled in liquid.

We thank the Reviewer’s suggestion and have clarified that the NTML library is presented in an arrayed format, enabling the detailed phenotypic profiling approach employed in this study (line 59-61, 78, 318).

3) The paper’s claim of novelty is justified for *S. aureus*; however, similar high-throughput, single-cell fluorescence microscopy approaches have been used to study bacteria-host cell interactions in other species. To more precisely define the novelty of this work, perhaps the authors could add a sentence like the following: “Where prior studies applied arrayed fluorescent promoter-based libraries in Gram-negative bacteria, our work applies high-resolution single-cell microscopy to an arrayed transposon library in a clinically important Gram-positive pathogen.”

Also please cite papers in other species that have used similar techniques, including:

- DOI: 10.1038/s41564-025-01953-5 :~3,000 arrayed fluorescent reporter strains of Salmonella to profile host–pathogen interactions in macrophages.
- DOI: 10.1038/nmeth895 :Comprehensive fluorescent reporter library in Escherichia coli for high-throughput based single cell imaging.

We appreciate the Reviewer’s suggestion to cite studies employing arrayed fluorescent promoter reporter libraries in Gram-negative bacteria to investigate transcriptional dynamics, including host–pathogen interactions (10.1038/s41564-025-01953-5). While we recognize the relevance of these studies, we consider they are not directly related to our work, which focuses on detailed phenotypic profiling of infection. Therefore, we have chosen not to include the suggested sentence or citations. We believe that the current presentation effectively conveys the novelty and significance of our findings within the context of *S. aureus* research.

4) As reviewers 2 and 3 have already pointed out, please continue to soften the language around “persistence”, as 48 hours post infection in vitro is not true persistence. Perhaps adding a statement in the Discussion noting that definitive persistence will require longer-term or in vivo models optimized for chronic infection would be useful here.

We have incorporated the Reviewer’s suggestion by adding a sentence to the Discussion section, as recommended (lines 332-333).

5) In lines 508-510, please add a little bit more technical detail into how THP1 differentiation was done, as this can vary widely across labs. For example, how long were cells differentiated for? Was there a resting period?

THP1 cells were differentiated by adding 50 ng/ml phorbol 12-myristate 13-acetate (PMA) at the time of plating and incubating for 72 h, without a resting period. This detail has been clarified in the Methods section of the manuscript (lines 513-515).

6) I really like that the authors added some data from clinical isolates in Fig. 7. Perhaps to visually highlight the diversity of these isolates, the authors could label in Fig 7 (or add a small schematic in the Fig) that these isolates come from patients with bacteremia, bone/joint infections, and infective endocarditis, respectively. This would visually make this conserved phenotype very striking.

We have updated Fig. 7 to clearly indicate the clinical origin of each isolate (bacteraemia, bone/joint infection, or infective endocarditis).